# The Impact of the Mini-batch Size on the Dynamics of SGD: Variance and Beyond

## Abstract

We study mini-batch stochastic gradient descent (SGD) dynamics under linear regression and deep linear networks by focusing on the variance of the gradients only given the initial weights and mini-batch size, which is the first study of this nature. In the linear regression case, we show that in each iteration the norm of the gradient is a decreasing function of the mini-batch size $b$ and thus the variance of the stochastic gradient estimator is a decreasing function of $b$. For deep neural networks with $L_2$ loss we show that the variance of the gradient is a polynomial in $1/b$. The results theoretically back the important intuition that smaller batch sizes yield larger variance of the stochastic gradients and lower loss function values which is a common believe among the researchers. The proof techniques exhibit a relationship between stochastic gradient estimators and initial weights, which is useful for further research on the dynamics of SGD. We empirically provide insights to our results on various datasets and commonly used deep network structures. We further discuss possible extensions of the approaches we build in studying the generalization ability of the deep learning models.

## 1    Introduction

Deep learning models have achieved great success in a variety of tasks including natural language processing, computer vision, and reinforcement learning (Goodfellow et al., 2016). Despite their practical success, there are only limited studies of the theoretical properties of deep learning; see survey papers (Sun, 2019; Fan et al., 2019) and references therein. The general problem underlying deep learning models is to optimize (minimize) a loss function, defined by the deviation of model predictions on data samples from the corresponding true labels. The prevailing method to train deep learning models is the mini-batch stochastic gradient descent algorithm and its variants (Bottou, 1998; Bottou et al., 2018). SGD updates model parameters by calculating a stochastic approximation of the full gradient of the loss function, based on a random selected subset of the training samples called a mini-batch.

It is well-accepted that selecting a large mini-batch size reduces the training time of deep learning models, as computation on large mini-batches can be better parallelized on processing units. For example, Goyal et al. (2017) scale ResNet-50 (He et al., 2016) from a mini-batch size of 256 images and training time of 29 hours, to a larger mini-batch size of 8,192 images. Their training achieves the same level of accuracy while reducing the training time to one hour. However, noted by many researchers, larger mini-batch sizes suffer from a worse generalization ability (LeCun et al., 2012; Keskar et al., 2017). Therefore, many efforts have been made to develop specialized training procedures that achieve good generalization using large mini-batch sizes (Hoffer et al., 2017; Goyal et al., 2017). Smaller batch sizes have the advantage of allegedly offering better generalization (at the expense of a higher training time).

The focus of this study is on the behavior of SGD subject to the conditions on the initial point. This is different from previous results which analyze SGD via stringing one-step recursions together. The dynamics of SGD are not comparable if we merely consider the one-step behavior, as the model parameters change iteration by iteration. Therefore, fixing the initial weights and the learning rate can give us a fair view of the impact of different mini-batch sizes on the dynamics of SGD. We hypothesize that, given the same initial point, smaller sizes lead to lower training loss and, unfortunately,

decrease stability of the algorithm on average. The latter follows from the fact that the smaller is the batch size, more stochasticity and volatility is introduced. After all, if the batch size equals to the number of samples, there is no stochasticity in the algorithm. To this end, we conjecture that the variance of the gradient in each iteration is a decreasing function of the mini-batch size. The conjecture is the focus of the work herein.

Variance correlates to many other important properties of SGD dynamics. For example, there is substantial work on variance reduction methods (Johnson & Zhang, 2013; Allen-Zhu & Hazan, 2016; Wang et al., 2013) which show great success on improving the convergence rate by controlling the variance of the stochastic gradients. Mini-batch size is also a key factor deciding the performance of SGD. Some research focuses on how to choose an optimal mini-batch size based on different criteria (Smith & Le, 2017; Gower et al., 2019). However, these works make strong assumptions on the loss function properties (strong or point or quasi convexity, or constant variance near stationary points) or about the formulation of the SGD algorithm (continuous time interpretation by means of differential equations). The statements are approximate in nature and thus not mathematical claims. The theoretical results regarding the relationship between the mini-batch size and the variance (and other performances, like loss and generalization ability) of the SGD algorithm applied to general machine learning models are still missing. The work herein partially addresses this gap by showing the impact of the mini-batch size on the variance of gradients in SGD. We further discuss possible extensions of the approaches we build in studying the generalization ability.

We are able to prove the hypothesis about variance in the convex linear regression case and to show significant progress in a deep linear neural network setting with samples based on a normal distribution. In this case we show that the variance is a polynomial in the reciprocal of the mini-batch size and that it is decreasing if the mini-batch size is larger than a threshold (further experiments reveal that this threshold can be as small as 2). The increased variance as the mini-batch size decreases should also intuitively imply convergence to lower training loss values and in turn better prediction and generalization ability (these relationships are yet to be confirmed analytically; but we provide empirical evidence to their validity).

The major contributions of this paper are as follows.

- For linear regression, we show that in each iteration the norm of any linear combination of sample-wise gradients is a decreasing function of the mini-batch size $b$ (Theorem 1). As a special case, the variance of the stochastic gradient estimator and the full gradient at the iterate in step $t$ are also decreasing functions of $b$ at any iteration step $t$ (Theorem 2). In addition, the proof provides a recursive relationship between the norm of gradients and the model parameters at each iteration (Lemma 2). This recursive relationship can be used to calculate any quantity related to the full/stochastic gradient or loss at any iteration with respect to the initial weights.

- For the deep linear neural network with $L_2$-loss and samples drawn from a normal distribution, we take two-layer linear network as an example and show that in each iteration step $t$ the trace of any product of the stochastic gradient estimators and weight matrices is a polynomial in $1/b$ with coefficients a sum of products of the initial weights (Theorem 3). As a special case, the variance of the stochastic gradient estimator is a polynomial in $1/b$ without the constant term (Theorem 4) and therefore it is a decreasing function of $b$ when $b$ is large enough (Theorem 5). The results and proof techniques can be easily extended to general deep linear networks. As a comparison, other papers that study theoretical properties of two-layer networks either fix one layer of the network, or assume the over-parameterized property of the model and they study convergence, while our paper makes no such assumptions on the model capacity. The proof also reveals the structure of the coefficients of the polynomial, and thus serving as a tool for future work on proving other properties of the stochastic gradient estimators.

- The proofs are involved and require several key ideas. The main one is to show a more general result than it is necessary in order to carry out the induction. The induction is on time step $t$. The key idea is to show a much more general result that lets us carry out induction. New concepts and definitions are introduced in order to handle the more general case. Along the way we show a result of general interest establishing expectation of several rank one matrices sampled from a normal distribution intertwined with constant matrices.

- We verify the theoretical results on various datasets and provide further understanding. We further empirically show that the results extend to other widely used network structures and hold for all choices of the mini-batch sizes. We also empirically verify that, on average, in each iteration the loss function value and the generalization ability (measured by the gap between accuracy on the training and test sets) are all decreasing functions of the mini-batch size.

In conclusion, we study the dynamics of SGD under linear regression and a two-layer linear network setting by focusing on the decreasing property of the variance of stochastic gradient estimators with respect to the mini-batch size. The proof techniques can also be used to derive other properties of the SGD dynamics in regard to the mini-batch size and initial weights. To the best of authors' knowledge, the work is the first one to theoretically study the impact of the mini-batch size on the variance of the gradient subject to the conditions on the initial weights, under mild assumptions on the network and the loss function. We support our theoretical results by experiments. We further experiment on other state-of-the-art deep learning models and datasets to empirically show the validity of the conjectures about the impact of mini-batch size on average loss, average accuracy and the generalization ability of the model.

The rest of the manuscript is structured as follows. In Section 2 we review the literature while in Section 3 we present the theoretical results on how mini-batch sizes impact the variance of stochastic gradient estimators, under different models including linear regression and deep linear networks. Section 4 introduces (part of) the experiments that verify our theorems and provide further insights into the impact of the mini-batch sizes on SGD performance. We defer the complete experimental details to Appendix A and the proofs of the theorems and other technical details to to Appendix B.

## 2 LITERATURE REVIEW

Stochastic gradient descent type methods are broadly used in machine learning (Bottou, 1991; LeCun et al., 1998; Bottou et al., 2018). The performance of SGD highly relies on the choice of the mini-batch size. It has been widely observed that choosing a large mini-batch size to train deep neural networks appears to deteriorate generalization (LeCun et al., 2012). This phenomenon exists even if the models are trained without any budget or limits, until the loss function value ceases to improve (Keskar et al., 2017). One explanation for this phenomenon is that large mini-batch SGD produces "sharp" minima that generalize worse (Hochreiter & Schmidhuber, 1997; Keskar et al., 2017). Specialized training procedures to achieve good performance with large mini-batch sizes have also been proposed (Hoffer et al., 2017; Goyal et al., 2017).

It is well-known that SGD has a slow asymptotic rate of convergence due to its inherent variance (Johnson & Zhang, 2013). Variants of SGD that can reduce the variance of the stochastic gradient estimator, which yield faster convergence, have also been suggested. The use of the information of full gradients to provide variance control for stochastic gradients is addressed in Johnson & Zhang (2013); Roux et al. (2012); Shalev-Shwartz & Zhang (2013). The works in Lei et al. (2017); Li et al. (2014); Schmidt et al. (2017) further improve the efficiency and complexity of the algorithm by carefully controling the variance.

There is prior work focusing on studying the dynamics of SGD. Neelakantan et al. (2015) propose to add isotropic white noise to the full gradient to study the "structured" variance. The works in Li et al. (2017); Mandt et al. (2017); Jastrzebski et al. (2017) connect SGD with stochastic differential equations to explain the property of converged minima and generalization ability of the model. Smith & Le (2017) propose an "optimal" mini-batch size which maximizes the test set accuracy by a Bayesian approach. The Stochastic Gradient Langevin Dynamics (SGLD, a variant of SGD) algorithm for non-convex optimization is studied in Zhang et al. (2017); Mou et al. (2018).

In most of the prior work about the convergence of SGD, it is assumed that the variance of stochastic gradient estimators is upper-bounded by a linear function of the norm of the full gradient, e.g. Assumption 4.3 in Bottou et al. (2018). Gower et al. (2019) give more precise bounds of the variance under different sampling methods and Khaled & Richtárik (2020) extend them to smooth non-convex regime. These bounds are still dependent on the model parameters at the corresponding iteration. To the best of the authors' knowledge, there is no existing result which represents the variance of

stochastic gradient estimators only using the initial weights and the mini-batch size. This paper partially solves this problem.

## 3 ANALYSIS

Mini-batch SGD is a lighter-weight version of gradient descent. Suppose that we are given a loss function $L(w)$ where $w$ is the collection (vector, matrix, or tensor) of all model parameters. At each iteration $t$, instead of computing the full gradient $\nabla_w L(w_t)$, SGD randomly samples a mini-batch set $\mathcal{B}_t$ that consists of $b = |\mathcal{B}_t|$ training instances and sets $w_{t+1} \leftarrow w_t - \alpha_t \nabla_w L_{\mathcal{B}_t}(w_t)$, where the positive scalar $\alpha_t$ is the learning rate (or step size) and $\nabla_w L_{\mathcal{B}_t}(w_t)$ denotes the stochastic gradient estimator based on mini-batch $\mathcal{B}_t$.

An important property of the stochastic gradient estimator $\nabla_w L_{\mathcal{B}_t}(w_t)$ is that it is an unbiased estimator, i.e. $\mathbb{E}\nabla_w L_{\mathcal{B}_t}(w_t) = \nabla_w L(w_t)$, where the expectation is taken over all possible choices of mini-batch $\mathcal{B}_t$. However, it is unclear what is the value of $\text{var}\left(\nabla_w L_{\mathcal{B}_t}(w_t)\right) \triangleq \mathbb{E}\left\|\nabla_w L_{\mathcal{B}_t}(w_t)\right\|^2 - \left\|\mathbb{E}\nabla_w L_{\mathcal{B}_t}(w_t)\right\|^2$. Intuitively, we should have $\text{var}\left(\nabla_w L_{\mathcal{B}_t}(w_t)\right) \propto \frac{n^2}{b}\text{var}\left(\nabla_w L(w_t)\right)$, where $n$ is the number of training samples and stochasticity on the right-hand side comes from mini-batch samples behind $w_t$ (Smith & Le, 2017; Gower et al., 2019). However, even the quantities $\nabla_w L(w_t)$ and $\text{var}\left(\nabla_w L(w_t)\right)$ are still challenging to compute as we do not have direct formulas of their precise values. Besides, as we choose different $b$'s, their values are not comparable as we end up with different $w_t$'s.

A plausible idea to address these issues is to represent $\mathbb{E}\nabla_w L_{\mathcal{B}_t}(w_t)$ and $\text{var}\left(\nabla_w L_{\mathcal{B}_t}(w_t)\right)$ using the fixed and known quantities $w_0, b, t$, and $\alpha_t$. In this way, we can further discover the properties, like decreasing with respect to $b$, of $\mathbb{E}\nabla_w L_{\mathcal{B}_t}(w_t)$ and $\text{var}\left(\nabla_w L_{\mathcal{B}_t}(w_t)\right)$. The biggest challenge is how to connect the quantities in iteration $t$ with those of iteration 0. This is similar to discovering the properties of a stochastic differential equation at time $t$ given only the dynamics of the stochastic differential equation and the initial point.

In this section, we address these questions under two settings: linear regression and a deep linear network. In Section 3.1 with a linear regression setting, we provide explicit formulas for calculating any norm of the linear combination of sample-wise gradients. We therefore show that the $\text{var}\left(\nabla_w L_{\mathcal{B}_t}(w_t)\right)$ is a decreasing function of the mini-batch size $b$. In Section 3.2 with a deep linear network setting and samples drawn from a normal distribution, we show that any trace of the product of weight matrices and stochastic gradient estimators is a polynomial in $1/b$ with finite degree. We further prove that $\text{var}\left(\nabla_w L_{\mathcal{B}_t}(w_t)\right)$ is a decreasing function of the mini-batch size $b > b_0$ for some constant $b_0$.

For a random matrix $M$, we define $\text{var}\left(M\right) \triangleq \mathbb{E}\left\|\text{vec}(M)\right\|^2 - \left\|\mathbb{E}\text{vec}(M)\right\|^2$ where $\text{vec}(M)$ denotes the vectorization of matrix $M$. We denote $[m:n] \triangleq \{m, m+1, \ldots, n\}$ if $m \leqslant n$, and $\varnothing$ otherwise. We use $[n] \triangleq [1:n]$ as an abbreviation. For clarity, we use the superscript $b$ to distinguish the variables with different choices of the mini-batch size $b$. In each iteration $t$, we use $\mathcal{B}_t^b$ to denote the batch of samples (or sample indices) to calculate the stochastic gradient. We denote by $\mathcal{F}_t^b$ the filtration of information before calculating the stochastic gradient in the $t$-th iteration, i.e. $\mathcal{F}_t^b \triangleq \left\{w_0, \mathcal{B}_0^b, \ldots, \mathcal{B}_{t-1}^b\right\}$.

### 3.1 LINEAR REGRESSION

In this subsection, we discuss the dynamics of SGD applied in linear regression. Given data points $(x_1, y_1), \cdots, (x_n, y_n)$, where $x_i \in \mathbb{R}^p$ and $y_i \in \mathbb{R}$, we define the loss function to be $L(w) = \frac{1}{n}\sum_{i=1}^n L_i(w) = \frac{1}{n}\sum_{i=1}^n \frac{1}{2}\left(w^T x_i - y_i\right)^2$, where $w \in \mathbb{R}^p$ are the model parameters. We consider minimizing $L(w)$ by mini-batch SGD. Note that the bias term in the general linear regression models is omitted, however, adding the bias term does not change the result of this section. Formally, we first choose a mini-batch size $b$ and initial weights $w_0$. In each iteration $t$, we sample $\mathcal{B}_t^b$, a subset of $[n]$ with cardinality $b$, and update the parameters by $w_{t+1}^b = w_t^b - \alpha_t g_t^b$, where $g_t^b = \frac{1}{b}\sum_{i \in \mathcal{B}_t^b} \nabla L_i\left(w_t^b\right)$.

We first show the relationship between the variance of stochastic gradient $g_t^b$ and the full gradient $\nabla L\left(w_t^b\right)$ and sample-wise gradient $\nabla L_i\left(w_t^b\right), i \in [n]$, derived by considering all possible choices

of the mini-batch $\mathcal{B}_t^b$. Readers should note that Lemma 1 actually holds for all models with $L_2$-loss, not merely linear regression (since in the proof we do not need to know the explicit form of $L_i(w)$).

**Lemma 1.** *Let $c_b \triangleq \frac{n-b}{b(n-1)} \geqslant 0$. For any matrix $A \in \mathbb{R}^{p \times p}$ we have* $\mathsf{var}\left(Ag_t^b \middle| \mathcal{F}_t^b\right) = \mathbb{E}\left[\left\|Ag_t^b\right\|^2 \middle| \mathcal{F}_t^b\right] - \left\|A\nabla L\left(w_t^b\right)\right\|^2 = c_b \left(\frac{1}{n}\sum_{i=1}^n \left\|A\nabla L_i\left(w_t^b\right)\right\|^2 - \left\|A\nabla L\left(w_t^b\right)\right\|^2\right).$

Lemma 1 provides a bridge to connect the norm and variance of $g_t^b$ with sample-wise gradients $\nabla L_i\left(w_t^b\right), i \in [n]$. Therefore, if we can further discover the properties of $\nabla L_i\left(w_t^b\right), i \in [n]$, we are able to calculate the variance of $g_t^b$. Lemma 2 addresses this problem by showing the relationship between any linear combination of $\nabla L_i\left(w_t^b\right)$'s and $\nabla L_i\left(w_{t-1}^b\right)$'s.

**Lemma 2.** *For any set of square matrices $\{A_1, \cdots, A_n\} \in \mathbb{R}^{p \times p}$, if we denote $A = \sum_{i=1}^n A_i x_i x_i^T$, then we have* $\mathbb{E}\left[\left\|\sum_{i=1}^n A_i \nabla L_i\left(w_{t+1}^b\right)\right\|^2 \middle| \mathcal{F}_0\right] = \mathbb{E}\left[\left\|\sum_{i=1}^n B_i \nabla L_i\left(w_t^b\right)\right\|^2 \middle| \mathcal{F}_0\right] + \frac{\alpha_t^2 c_b}{n^2}\sum_{k=1}^n \sum_{l=1}^n \mathbb{E}\left[\left\|\sum_{i=1}^n B_i^{kl} \nabla L_i\left(w_t^b\right)\right\|^2 \middle| \mathcal{F}_0\right]$, *where $B_i = A_i - \frac{\alpha_t}{n}A$; $B_i^{kl} = A$ if $i = k, i \neq l$, $B_i^{kl} = A$ if $i = l, i \neq k$, and $B_i^{kl}$ equals the zero matrix, otherwise.*

Lemma 2 provides the tool to reduce the iteration $t$ by one. Therefore, we can easily use it to recursively calculate the norm of any linear combinations of the sample-wise gradients, for all iterations $t$. Combining the fact that $c_b$ is a decreasing function of $b$, we are able to show Theorem 1.

**Theorem 1.** *For any $t \in \mathbb{N}$ and any matrices $A_i \in \mathbb{R}^{p \times p}, i \in [n]$, $\mathbb{E}\left[\left\|\sum_{i=1}^n A_i \nabla L_i\left(w_t^b\right)\right\|^2 \middle| \mathcal{F}_0\right]$ is a decreasing function of $b$ for $b \in [n]$.*

Theorem 1 states that the norm of any linear combinations of the sample-wise gradients is a decreasing function of $b$. Combining Lemma 1 which connects the variance of $g_t^b$ with the linear combination of $\nabla L_i\left(w_t^b\right)$'s, and the fact that $\nabla L\left(w_t^b\right) = \frac{1}{n}\sum_{i=1}^n \nabla L_i\left(w_t^b\right)$, we have Theorem 2.

**Theorem 2.** *Fixing initial weights $w_0$, both $\mathsf{var}\left(Bg_t^b \middle| \mathcal{F}_0\right)$ and $\mathsf{var}\left(B\nabla L\left(w_t^b\right) \middle| \mathcal{F}_0\right)$ are decreasing functions of mini-batch size $b$ for all $b \in [n]$, $t \in \mathbb{N}$, and all square matrices $B \in \mathbb{R}^{p \times p}$.*

As a special case, Corollary 1 guarantees that the variance of the stochastic gradient estimator is a decreasing function of $b$.

**Corollary 1.** *Fixing initial weights $w_0$, both $\mathsf{var}\left(g_t^b \middle| \mathcal{F}_0\right)$ and $\mathsf{var}\left(\nabla L\left(w_t^b\right) \middle| \mathcal{F}_0\right)$ are decreasing functions of mini-batch size $b$ for all $b \in [n]$ and $t \in \mathbb{N}$.*

In conclusion, we provide a framework for calculating the explicit value of variance of the stochastic gradient estimators and the norm of any linear combination of sample-wise gradients. We further show that the variance of both the full gradient and the stochastic gradient estimator are a decreasing function of the mini-batch size $b$. Readers should note that the framework here is not limited to showing the decreasing property of the variance, but can also be used in many other circumstance. For example, we can use Lemma 2 to induct on $t$ and easily show that $\mathbb{E}\left[\left\|\sum_{i=1}^n A_i \nabla L_i\left(w_t^b\right)\right\|^2 \middle| \mathcal{F}_0\right]$ is a polynomial of $\frac{1}{b}$ with degree at most $t$ and estimate the coefficients therein.

### 3.2 DEEP LINEAR NETWORKS WITH ONLINE SETTING

In this section, we study the dynamics of SGD on deep linear networks. We take the two-layer linear network as an example while the results and proofs can be easily extended to deep linear network with any depth (see Appendix B.3 for more details). We consider the population loss $\mathcal{L}(w) = \mathbb{E}_{x \sim \mathcal{N}(0, I_p)}\left[\frac{1}{2}\left\|W_2 W_1 x - W_2^* W_1^* x\right\|^2\right]$ under the teacher-student learning framework (Hinton et al., 2015) with $w = (W_1, W_2)$ a tuple of two matrices. Here $W_1 \in \mathbb{R}^{p_1 \times p}$ and $W_2 \in \mathbb{R}^{p_2 \times p_1}$ are parameter matrices of the student network and $W_1^*$ and $W_2^*$ are the fixed ground-truth parameters of the teacher network. We use online SGD to minimize the population loss $\mathcal{L}(w)$. Formally, we first choose a mini-batch size $b$ and initial weight matrices $\{W_{0,1}, W_{0,2}\}$. In each iteration $t$, we draw $b$ independent and identically distributed samples $x_{t,i}, i \in [b]$ from $\mathcal{N}(0, I_p)$ to form the mini-batch $\mathcal{B}_t^b$ and update the weight matrices by $W_{t+1,1}^b = W_{t,1}^b - \alpha_t g_{t,1}^b$ and

$W_{t+1,2}^b = W_{t,2}^b - \alpha_t g_{t,2}^b$, where

$$g_{t,1}^b = \frac{1}{b}\sum_{i=1}^b \nabla_{W_{t,1}^b}\left(\frac{1}{2}\left\|W_{t,2}^b W_{t,1}^b x_{t,i} - W_2^* W_1^* x_{t,i}\right\|^2\right) = \frac{1}{b}\sum_{i=1}^b {W_{t,2}^b}^T\left(W_{t,2}^b W_{t,1}^b - W_2^* W_1^*\right) x_{t,i} x_{t,i}^T, \quad (1)$$

$$g_{t,2}^b = \frac{1}{b}\sum_{i=1}^b \nabla_{W_{t,2}^b}\left(\frac{1}{2}\left\|W_{t,2}^b W_{t,1}^b x_{t,i} - W_2^* W_1^* x_{t,i}\right\|^2\right) = \frac{1}{b}\sum_{i=1}^b \left(W_{t,2}^b W_{t,1}^b - W_2^* W_1^*\right) x_{t,i} x_{t,i}^T {W_{t,1}^b}^T. \quad (2)$$

The derivation follows from the formulas in Petersen & Pedersen (2012). In the following, we use $\mathcal{W}_t^b = W_{t,2}^b W_{t,1}^b - W_2^* W_1^*$ to denote the gap between the product of model weights and ground-truth weights.

For ease of developing our proofs, we first introduce the definition of a *multiplicative term* in Definition 1. Intuitively, a multiplicative term is a matrix which equals to the product of its parameter matrices and constant matrices (and their transpose). The degree of a matrix $A$ in a multiplicative term $M$ is the number of appearance of $A$ and $A^T$ in $M$. The degree of $M$ is exactly the number of appearances of all weight matrices in $M$.

**Definition 1.** For any set of matrices $\mathcal{S}$, we denote $\bar{\mathcal{S}} = \mathcal{S} \cup \{M^T : M \in \mathcal{S}\}$. Given a set of parameter matrices $\mathcal{X} = \{X_1, X_2, \cdots, X_{n_v}\}$ and constant matrices $\mathcal{C} = \{C_1, C_2, \cdots, C_{n_c}\}$, we say that a matrix $M$ is *a multiplicative term of parameter matrices $\mathcal{X}$ and constant matrices $\mathcal{C}$* if it can be written in the form of $M = M(\mathcal{X},\mathcal{C}) = \prod_{i=1}^k A_i$, where $A_i \in \bar{\mathcal{X}} \cup \bar{\mathcal{C}}$. We write $\deg(X_j; M) = \sum_{i=1}^k \left(\mathbb{1}\{X_j = A_i\} + \mathbb{1}\{X_j = A_i^T\}\right), j \in [n_v]$ as the degree of parameter matrix $X_j$ in $M$, $\deg(C_j; M) = \sum_{i=1}^k \left(\mathbb{1}\{C_j = A_i\} + \mathbb{1}\{C_j = A_i^T\}\right), j \in [n_c]$ as the degree of constant matrix $C_j$ in $M$, and $\deg(M) = \sum_{i=1}^k \mathbb{1}\{A_i \in \bar{\mathcal{X}}\} = \sum_{j=1}^{n_v}\deg(X_j; M)$ as the total degree of the parameter matrices of $M$.

As pointed out in the Section 1, the difficulty of studying the dynamics of SGD is how to connect the quantities in iteration $t$ with fixed variables, like initial weights $W_{0,1}, W_{0,2}$ and mini-batch size $b$. We overcome this challenge by carefully calculating the relationship between $g_{t+1,i}^b$ and $g_{t,i}^b, i = 1, 2$ so that we can reduce the iteration $t$ step by step. With the help of Lemmas 8 and 9 in Appendix B.2, we can represent $g_{t+1,i}^b, i = 1, 2$ using multiplicative terms of $g_{t,i}^b, i = 1, 2$ and some other constant matrices. Theorem 3 precisely gives the representation in the form of a polynomial of $\frac{1}{b}$ and the coefficients as the sum of multiplicative terms of parameter matrices $\{W_{0,1}^b, W_{0,2}^b\}$ and constant matrices $\{W_1^*, W_2^*\}$.

**Theorem 3.** *Given $t \geqslant 0$, for any multiplicative terms $M_i, i \in [0:m]$ of parameter matrices $\{g_{t,1}^b, g_{t,2}^b\}$ and constant matrices $\{W_{t,1}^b, W_{t,2}^b, W_1^*, W_2^*\}$ with degree $d_i$, respectively, we denote $M = \prod_{i=1}^m \operatorname{tr}(M_i) M_0$, $d = \sum_{i=0}^m d_i$ and $d' = \sum_{i=0}^m \left(\deg\left(W_{t,1}^b; M_i\right) + \deg(W_{t,2}^b; M_i)\right)$. There exists a set of multiplicative terms $\left\{M_{ij}^k, i \in [m_k], j \in [0:m_{ki}], k \in [0:q]\right\}$ of parameter matrices $\{W_{0,1}^b, W_{0,2}^b\}$ and constant matrices $\{W_1^*, W_2^*\}$ such that $\mathbb{E}[M|\mathcal{F}_0] = N_0 + N_1\frac{1}{b} + \cdots + N_q\frac{1}{b^q}$, where $N_k = \sum_{i=1}^{m_k}\prod_{j=1}^{m_{ki}}\operatorname{tr}\left(M_{ij}^k\right) M_{i0}^k, k \in [0:q]$. Here $m_k, m_{ki}$ and $q \leqslant \frac{1}{2}(3^{t+1}-1)d + \frac{1}{2}(3^t-1)d'$ are constants independent of $b$, and $\sum_{j=0}^{m_{ki}}\deg\left(M_{ij}^k\right) \leqslant 3^t(3d + d')$.*

As a special case of Theorem 3, Theorem 4 shows that the variance of the stochastic gradient estimators is also a polynomial of $\frac{1}{b}$ but with no constant term. This backs the important intuition that the variance is approximately inversely proportional to the mini-batch size $b$. Besides, note that if we consider $b \to \infty$, intuitively we should have $\operatorname{var}\left(g_{t,i}^b\middle|\mathcal{F}_0\right) \to 0, i = 1, 2$. This observation aligns with the statement of Theorem 4.

**Theorem 4.** *Given $t \geqslant 0$, value $\operatorname{var}\left(g_{t,i}^b\middle|\mathcal{F}_0\right), i = 1, 2$ can be written as a polynomial of $\frac{1}{b}$ with degree at most $2 \cdot 3^t$ with no constant term. Formally, we have $\operatorname{var}\left(g_{t,i}^b\middle|\mathcal{F}_0\right) = \beta_1\frac{1}{b} + \cdots + \beta_r\frac{1}{b^r}$, where $r \leqslant 2 \cdot 3^{t+1}$ and each $\beta_i$ is a constant independent of $b$.*

One should note that the polynomial representation of $\operatorname{var}\left(g_{t,i}^b\middle|\mathcal{F}_0\right), i = 1, 2$ does not have the constant term. Therefore, to show the that the variance is a decreasing function of $b$, we only need to show that the leading coefficient $\beta_1$ is non-negative. This is guaranteed by the fact that variance is always non-negative. We therefore have Theorem 5.

**Theorem 5.** *Given $t \in \mathbb{N}$, there exists a constant $b_0$ such that for all $b \geqslant b_0$ function* $\text{var}\left(g_{t,i}^b | \mathcal{F}_0\right), i = 1, 2$ *is a decreasing function of $b$.*

The constant $b_0$ is the largest root of the equation $\beta_1 b^{r-1} + \beta_2 b^{r-2} + \cdots + \beta_r = 0$. See the proof of Theorem 5 in Appendix B.2 for more details. Although we cannot calculate the precise value of $b_0$, we verify that $b_0$ is smaller than 1 in many experiments. From the proofs we conclude that the scale of each $\beta_i$ is of the order $\mathcal{O}\left(\|M\|\right)$, where $M$ is a multiplicative term of parameter matrices $\{W_{0,1}, W_{0,2}, W_1^*, W_2^*\}$ and constant matrix $\varnothing$ with degree $2 \cdot 3^{t+1}$.

Unlike the linear regression setting where we can iteratively calculate the variance by Lemma 2, the closed form expressions for the variance of the stochastic gradients in the deep linear network setting are much harder to calculate. However, we are able to iteratively deducing $t$ one by one and provide a polynomial representation for any multiplicative terms of parameter matrices $\{g_{t,i}^b, W_{t,i}^b, i = 1, 2\}$ and constant matrices $\{W_1^*, W_2^*\}$ using only the initial weights $W_{0,1}, W_{0,2}$ and the mini-batch size $b$. As we further study the polynomial representation of $\text{var}\left(g_{t,i}^b | \mathcal{F}_0\right), i = 1, 2$, we are also able to show the decreasing property of the variance of stochastic gradient estimators with respect to $b$.

## 4 EXPERIMENTS

In this section, we present numerical results to support the theorems in Section 3 and provide further insights into the impact of the mini-batch size on the dynamics of SGD. The experiments are conducted on four datasets and models that are relatively small due to the computational cost of using large models and datasets. We only report the results on the MNIST dataset here due to the limited space. A complete empirical study is deferred in Appendix A.

For all experiments, we perform mini-batch SGD multiple times starting from the same initial weights and following the same choice of the learning rates and other hyper-parameters, if applicable. This enables us to calculate the variance of the gradient estimators and other statistics in each iteration, where the randomness comes only from different samples of SGD.

### 4.1 RESULTS ON MNIST DATASET

The MNIST dataset is to recognize digits in handwritten images of digits. We use all 60,000 training samples and 10,000 validation samples of MNIST. We build a three-layer fully connected neural network with 1024, 512 and 10 neurons in each layer. For the two hidden layers, we use the ReLU activation function. The last layer is the softmax layer which gives the prediction probabilities for the 10 digits. We use mini-batch SGD to optimize the cross-entropy loss of the model. The model deviates from our analytical setting since it has non-linear activations, it has the cross-entropy loss function (instead of $L_2$), and empirical loss (as opposed to population). MNIST is selected due to its fast training and popularity in deep learning experiments. The goal is to verify the results in this different setting and to back up our hypotheses.

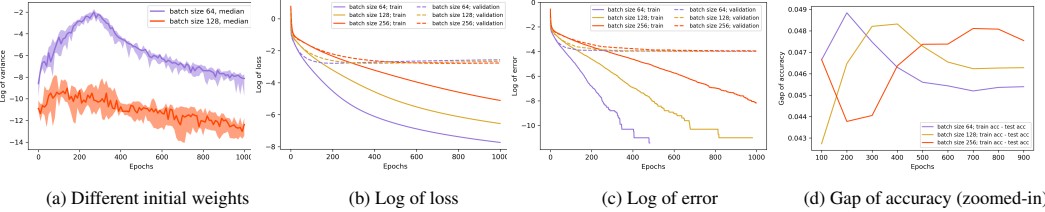

| (a) Different initial weights | (b) Log of loss | (c) Log of error | (d) Gap of accuracy (zoomed-in) |

Figure 1: Experimental results for the MNIST dataset. (a) The median, min, and max of the log of variance of the stochastic gradient estimators for two different mini-batch sizes (distinguished by colors) and five different initial weights. The solid lines show the median of all five initial weights while the highlighted regions show the min and max of the log of variance. (b) The log of the training and validation loss vs epochs. (c) The log of training and validation error vs epochs. Here error is defined as one minus predicting accuracy. The plot does not show the epochs if error equals to zero. (d) The gap of accuracy on training and test sets vs epochs starting from epoch 100.

As shown in Figure 1(a), we run SGD with two batch sizes 64 and 128 on five different initial weights with 50 runs for each initial point. This plot shows that, even the smallest value of the variance among the five different initial weights with a mini-batch size of 64, is still larger than the largest variance of mini-batch size 128. We observe that the sensitivity to the initial weights is not large. This plot also empirically verifies our conjecture in the introduction that the variance of the stochastic gradient estimators is a decreasing function of the mini-batch size, for all iterations of SGD in a general deep learning model.

In addition, we also conjecture that there exists the decreasing property for the expected loss, error and the generalization ability with respect to the mini-batch size. Figure 1(b) shows that the expected loss (again, randomness comes from different runs of SGD through the different mini-batches with the same initial weights and learning rates) on the training set is a decreasing function of $b$. However, this decreasing property does not hold on the validation set when the loss tends to be stable or increasing, in other words, the model starts to be over-fitting. We hypothesize that this is because the learned weights start to bounce around a local minimum when the model is over-fitting. As the larger mini-batch size brings smaller variance, the weights are closer to the local minimum found by SGD, and therefore yield a smaller loss function value. Figure 1(c) shows that both the expected error on training and validation sets are decreasing functions of $b$.

Figure 1(d) exhibits a relationship between the model's generalization ability and the mini-batch size. As suggested by Simard et al. (2013), we build a test set by distorting the 10,000 images of the validation set. The prediction accuracy is obtained on both training and test sets and we calculate the gap between these two accuracies every 100 epochs. We use this gap to measure the model generalization ability (the smaller the better). Figure 1(d) shows that the gap is an increasing function of $b$ starting at epoch 500, which partially aligns with our conjecture regarding the relationship between the generalization ability and the mini-batch size. We test this on multiple choices of the hyper-parameters which control the degree of distortion in the test set and this pattern remains clear.

## 5 SUMMARY AND FUTURE WORK

We examine the impact of the mini-batch size on the dynamics of SGD. Our focus is on the variance of stochastic gradient estimators. For linear regression and a two-layer linear network, we are able to theoretically prove that the variance conjecture holds. We further experiment on multiple models and datasets to verify our claims and their applicability to practical settings. Besides, we also empirically address the conjectures about the expected loss and the generalization ability.

A challenging research direction is to theoretically investigate the impact of the mini-batch size on the generalization ability. There are existing works studying the relationship between the variance of the stochastic gradients and the generalization ability (Gorbunov et al., 2020; Meng et al., 2016). Together with the tools developed herein, it would be possible to bridge the mini-batch size with the generalization ability of a neural network. We can further choose an optimal mini-batch size which minimizes the generalization ability by solving the polynomial equation if we have more precise estimations of the coefficients.

Another appealing direction is using our variance estimations to develop better variance reduction methods. As a results, the upper-bound of the variance decides the convergent rate of these algorithms. Researchers usually assume a much larger upper-bound at each iteration, like a linear function of the norm of the full gradient. With the help of our techniques, we should calculate the variance more precisely and further improve the algorithms.

Further interesting work is to extend our techniques to more complicated and sophisticated networks. Although the underlying model of this paper corresponds to deep linear network networks, we are able to show a deeper relationship between the variance and the mini-batch size, the polynomial in $1/b$, while the common knowledge is simply that the variance is proportional to $1/b$. The extension to other optimization algorithms, like Adam and Gradient Boosting Machines, are also very attractive. We hope our theoretical framework can serve as a tool for future research of this kind.

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

## A    EXPERIMENTS

In this section, we present numerical results to support the theorems in Section 3, to backup the hypotheses discussed in the introduction, and provide further insights into the impact of the mini-batch size on the dynamics of SGD. The experiments are conducted on four datasets and models that are relatively small due to the computational cost of using large models and datasets.

**Remark:** We cannot present the complete numerical results in the main paper due to the space limit. Therefore, we move the whole experimental section to Appendix. In order to keep a smooth reading, some of the content is overlapping with Section 4.

### A.1    DATASETS AND SETTINGS

For all experiments, we perform mini-batch SGD multiple times starting from the same initial weights and following the same choice of the learning rates and other hyper-parameters, if applicable. This enables us to calculate the variance of the gradient estimators and other statistics in each iteration, where the randomness comes only from different samples of SGD. The learning rate $\alpha_t$ is selected to be inversely proportional to iteration $t$, or fixed, depending on the task at hand.

All models are implemented using PyTorch version 1.4 (Paszke et al., 2019) and trained on NVIDIA 2080Ti/1080 GPUs. We have also tested several other random initial weights and ground-truth weights, and learning rates, and the results and conclusions are similar and not presented.

#### A.1.1    GRADUATE ADMISSION DATASET

The Graduate Admission dataset[1] (Acharya et al., 2019) is to predict the chance of a graduate admission using linear regression. The dataset contains 500 samples with 6 features and is normalized by mean and variance of each feature. This is a popular regression dataset with clean data. We build a linear regression model to predict the chance of acceptance (we include the intercept term in the model) and minimize the empirical $L_2$ loss using mini-batch SGD, as stated in Section 3.1.

For the experiment in Figure 2(a), we randomly select an initial weight vectors $w_0$ and run SGD for 2,000 iterations where it appears to converge. We record all statistics at every iteration. There are in total 1,000 runs behind each observation which yields a p-value lower than 0.05. As for Figure 2(b), we select 20 different $b$'s and run SGD from the same initial point for 40 iterations. There are in total of 200,000 runs to make sure the p-value of all statistics are lower than 0.05. In all experiments, the learning rate is chosen to be $\alpha_t = \frac{1}{2t}, t \in [2000]$ because this rate yields a theoretical convergence guaranteed (factor 1/2 has been fine tuned). The purpose of this experiment is to empirically study the rate of decrease of the variance. The theoretical study exhibited in Section 3.1 establishes the non-increasing property but it does not state anything about the rate of decrease.

#### A.1.2    SYNTHETIC DATASET

We build a synthetic dataset of standard normal samples to study the setting in Section 3.2. We fix the teacher network with 64 input neurons, 256 hidden neurons and 128 output neurons. We optimize the population $L_2$ loss by updating the two parameter matrices of the student network using online SGD, as stated in Section 3.2. In this case we have proved the functional form of the variance as a function of $b$ and show the decreasing property of the variance of the stochastic gradient estimators for large mini-batch sizes. However, we do not show the decreasing property for every $b$. With this experiment we confirm that the conjecture likely holds. In the experiment, we randomly select two initial weight matrices $W_{0,1}, W_{0,2}$ and the ground-truth weight matrices $W_1^*, W_2^*$. We run SGD for 1,000 iterations which appears to be a good number for convergence while there are 1,000 runs of SGD in total to again give a p-value below 0.05. We record all statistics at every iteration. The learning rate is chosen to be $\alpha_t = \frac{1}{10t}, t \in [1000]$ for the same reason as in the regression experiment.

---

[1]https://www.kaggle.com/mohansacharya/graduate-admissions

### A.1.3 MNIST DATASET

The MNIST dataset is to recognize digits in handwritten images of digits. We use all 60,000 training samples and 10,000 validation samples of MNIST. The images are normalized by mapping each entry to $[-1, 1]$. We build a three-layer fully connected neural network with 1024, 512 and 10 neurons in each layer. For the two hidden layers, we use the ReLU activation function. The last layer is the softmax layer which gives the prediction probabilities for the 10 digits. We use mini-batch SGD to optimize the cross-entropy loss of the model. The model deviates from our analytical setting since it has non-linear activations, it has the cross-entropy loss function (instead of $L_2$), and empirical loss (as opposed to population). MNIST is selected due to its fast training and popularity in deep learning experiments. The goal is to verify the results in this different setting and to back up our hypotheses.

We run SGD for 1,000 epochs on the training set which is enough for convergence. The learning rate is a constant set to $3 \cdot 10^{-3}$ (which has been tuned). For the experiment in Figure 5, there are in total 100 runs to give us the p-value below 0.05. For the experiment in Figure 4(a), we randomly select five different initial points and we have 50 runs for each initial point. For the experiment corresponding to Figure 4(b), we choose $\alpha = 8$ and $\sigma = 2$ as in Simard et al. (2013). The initial weights and other hyper-parameters are chosen to be the same as in Figure 5.

### A.1.4 YELP REVIEW DATASET

The Yelp Review dataset from the Yelp Dataset Challenge (Zhang et al., 2015) contains 1,569,264 samples of customer reviews with positive/negative sentiment labels. We use 10,000 samples as our training set and 1,000 samples as the validation set. We use XLNet (Yang et al., 2019) to perform sentiment classification on this dataset. Our XLNet has 6 layers, the hidden size of 384, and 12 attention heads. There are in total 35,493,122 parameters. We intentionally reduce the number of layers and hidden size of XLNet and select a relatively small size of the training and validation sets since training of XLNet is very time-consuming (Yang et al. (2019) train on 512 TPU v3 chips for 5.5 days) and we need to train the model for multiple runs. This setting allows us to train our model in several hours on a single GPU card. We train the model using the Adam weight decay optimizer, and some other techniques, as suggested in Table 8 of Yang et al. (2019). This dataset represents sequential data where we further consider the hypotheses.

We randomly select a set of initial parameters and run Adam with two different mini-batch sizes of 32 and 64. For computational tractability reasons, for each mini-batch size there are in total of 100 runs and each run corresponds to 20 epochs. We record the variance of the stochastic gradient, loss and accuracy in every step of Adam. The statistics reported in Figure 6 are averaged through each epoch. In all experiments, the learning rate is set to be $4 \cdot 10^{-5}$ and the $\epsilon$ parameter of Adam is set to be $10^{-8}$ (these two have been tuned). The stochastic gradients of all parameter matrices are clipped with threshold 1 in each iteration. We use the same setup for the learning rate warm-up strategy as suggested in Yang et al. (2019). The maximum sequence length is set to be 128 and we pad the sequences with length smaller than 128 with zeros.

## A.2 DISCUSSION

As observed in Figure 2(a), under the linear regression setting with the Graduate Admission dataset, the variance of the stochastic gradient estimators and full gradients are all strictly decreasing functions of $b$ for all iterations. This result verifies the theorems in Section 3.1. Figure 2(b) further studies the rate of decrease of the variance. From the proofs in Section 3.1 we see that $\mathrm{var}\left(g_t^b|\mathcal{F}_0\right)$ is a polynomial of $\frac{1}{b}$ with degree $t + 1$. Therefore, for every $t$, we can approximate this polynomial by sampling many different $b$'s and calculate the corresponding variances. We pick $b$ to cover all numbers that are either a power of 2 or multiple of 40 in $[2, 500]$ (there are a total of 21 such values) and fit a polynomial with degree 6 (an estimate from the analyses) at $t = 10, 20, 30, 40$. Figure 2(b) shows the fitted polynomials. As we observe, the value $\mathrm{var}\left(g_t^b|\mathcal{F}_0\right)$ (approximated by the value of the polynomial) is both decreasing with respect to the mini-batch size $b$ and iteration $t$. Further, the rate of decrease in $b$ is slower as the $b$ increasing. This provides a further insight into the dynamics of training a linear regression problem with SGD.

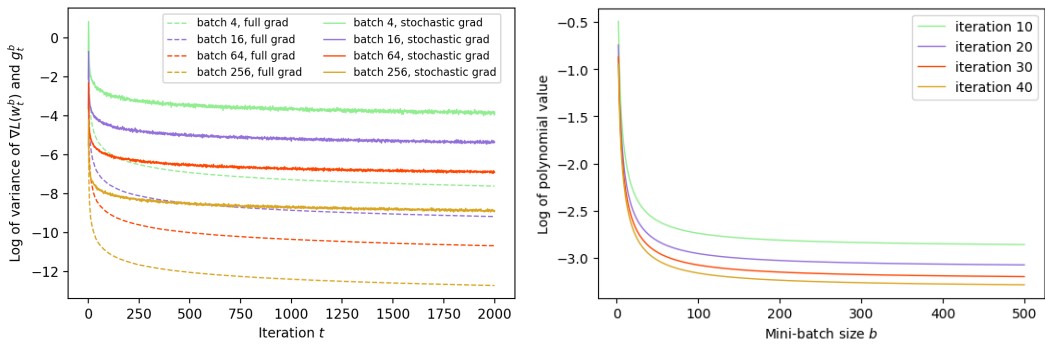

(a) Variance of stochastic gradients and full gradients

(b) Fitting polynomials of mini-batch size $b$

Figure 2: Experimental results for the Graduate Admission dataset. **Left:** $\log\left(\text{var}\left(g_t^b\big|\mathcal{F}_0\right)\right)$ and $\log\left(\text{var}\left(\nabla L(w_t^b)\big|\mathcal{F}_0\right)\right)$ vs iteration $t$ for 4 different mini-batch sizes. **Right:** The log of polynomial values when fitting polynomials on selected mini-batch sizes at certain iterations.

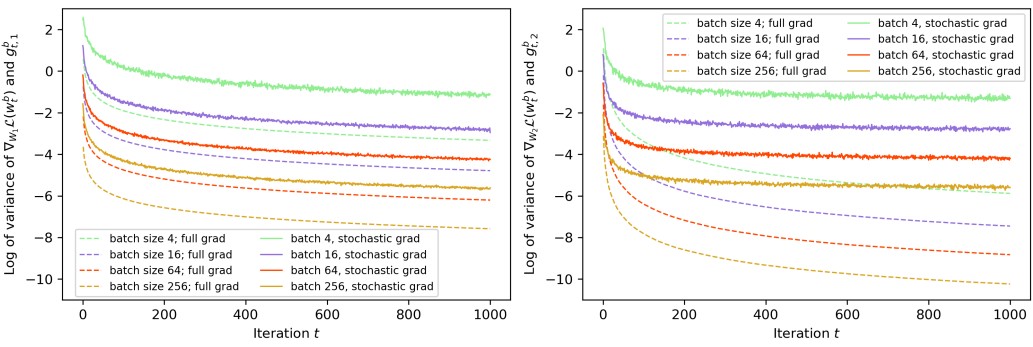

(a) Variance of gradients with respect to $W_1$

(b) Variance of gradients with respect to $W_2$

Figure 3: Experimental results for the Synthetic dataset. **Left:** $\log\left(\text{var}\left(g_{t,1}^b\big|\mathcal{F}_0\right)\right)$ and $\log\left(\text{var}\left(\nabla_{W_1}\mathcal{L}(W_{t,1}^b, W_{t,2}^b)\big|\mathcal{F}_0\right)\right)$ vs iteration $t$. **Right:** $\log\left(\text{var}\left(g_{t,2}^b\big|\mathcal{F}_0\right)\right)$ and $\log\left(\text{var}\left(\nabla_{W_2}\mathcal{L}(W_{t,1}^b, W_{t,2}^b)\big|\mathcal{F}_0\right)\right)$ vs iteration $t$.

Under the two-layer linear network setting with the synthetic dataset, Figure 3 verifies that the variance of the stochastic gradient estimators and full gradients are all strictly decreasing functions of $b$ for all iterations. This figure also empirically shows that the constant $b_0$ in Theorem 5 could be as small as $b_0 = 4$. In fact, we also experiment with the mini-batch size of 1 and 2, and the decreasing property remains to hold. We also test this on multiple choices of initial weights and learning rates and this pattern remains clear.

In aforementioned two experiments we use SGD in its original form by randomly sampling mini-batches. In deep learning with large-scale training data such a strategy is computationally prohibitive and thus samples are scanned in a cyclic order which implies fixed mini-batches are processed many times. Therefore, in the next two datasets we perform standard "epoch" based training to empirically study the remaining two hypotheses discussed in the introduction (decreasing loss and error as a function of $b$) and sensitivity with respect to the initial weights. Note that we are using cross-entropy loss in the MNIST dataset and the Adam optimizer in the Yelp dataset and thus these experiments do not meet all of the assumptions of the analysis in Section 3.

As shown in Figure 4(a), we run SGD with two batch sizes 64 and 128 on five different initial weights. This plot shows that, even the smallest value of the variance among the five different initial weights with a mini-batch size of 64, is still larger than the largest variance of mini-batch size 128. We observe that the sensitivity to the initial weights is not large. This plot also empirically verifies our conjecture in the introduction that the variance of the stochastic gradient estimators is

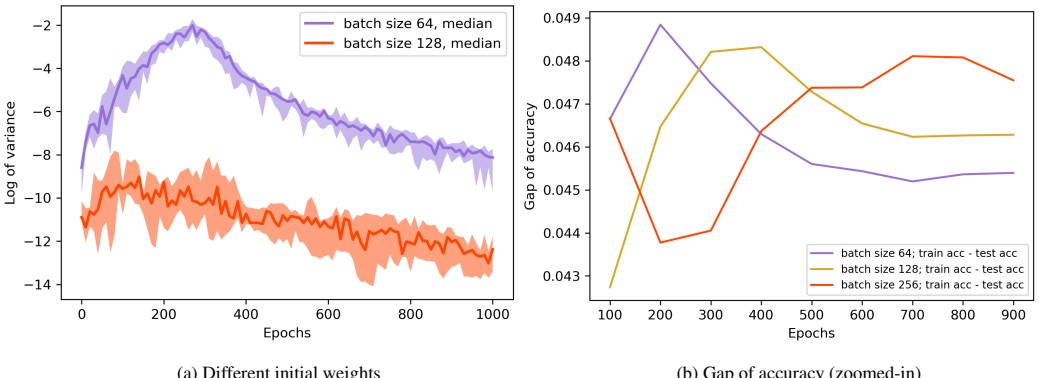

(a) Different initial weights

(b) Gap of accuracy (zoomed-in)

Figure 4: Experimental results for the MNIST dataset. **Left:** The median, min, and max of the log of variance of the stochastic gradient estimators for two different mini-batch sizes (distinguished by colors) and five different initial weights. The solid lines show the median of all five initial weights while the highlighted regions show the min and max of the log of variance. **Right:** The gap of accuracy on training and test sets vs epochs starting from epoch 100.

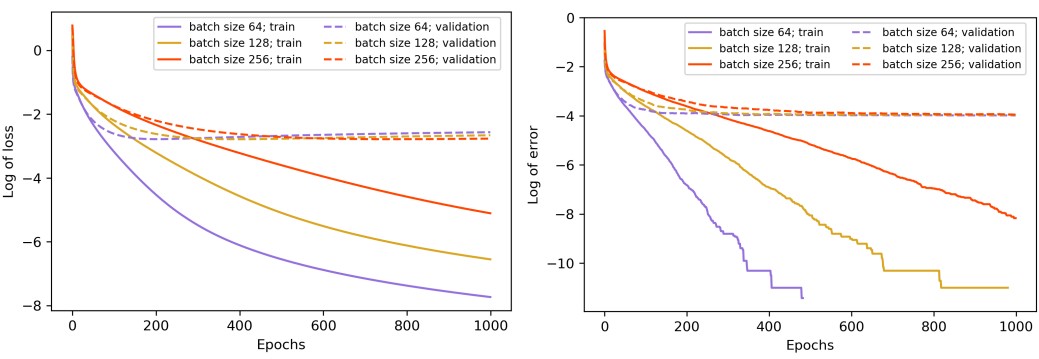

(a) Log of loss for training and validation sets

(b) Log of error for training and validation sets

Figure 5: Experimental results for the MNIST dataset. **Left:** The log of the training and validation loss vs epochs. **Right:** The log of training and validation error vs epochs. Here error is defined as one minus predicting accuracy. The plot does not show the epochs if error equals to zero.

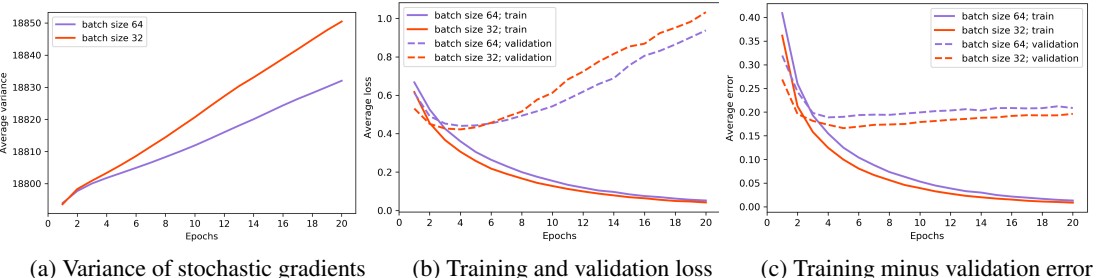

(a) Variance of stochastic gradients

(b) Training and validation loss

(c) Training minus validation error

Figure 6: Experimental results for the XLNet model on the Yelp dataset. **Left:** The variance of stochastic gradient estimators vs epochs. **Middle:** The training and validation loss vs epochs. **Right:** The training and validation error vs epochs.

a decreasing function of the mini-batch size, for all iterations of SGD in a general deep learning model.

In addition, we also conjecture that there exists the decreasing property for the expected loss, error and the generalization ability with respect to the mini-batch size. Figure 5(a) shows that the expected

loss (again, randomness comes from different runs of SGD through the different mini-batches with the same initial weights and learning rates) on the training set is a decreasing function of $b$. However, this decreasing property does not hold on the validation set when the loss tends to be stable or increasing, in other words, the model starts to be over-fitting. We hypothesize that this is because the learned weights start to bounce around a local minimum when the model is over-fitting. As the larger mini-batch size brings smaller variance, the weights are closer to the local minimum found by SGD, and therefore yield a smaller loss function value. Figure 5(b) shows that both the expected error on training and validation sets are decreasing functions of $b$.

Figure 4(b) exhibits a relationship between the model's generalization ability and the mini-batch size. As suggested by (Simard et al., 2013), we build a test set by distorting the 10,000 images of the validation set. The prediction accuracy is obtained on both training and test sets and we calculate the gap between these two accuracies every 100 epochs. We use this gap to measure the model generalization ability (the smaller the better). Figure 4(b) shows that the gap is an increasing function of $b$ starting at epoch 500, which partially aligns with our conjecture regarding the relationship between the generalization ability and the mini-batch size. We also test this on multiple choices of the hyper-parameters which control the degree of distortion in the test set and this pattern remains clear.

Figure 6 shows the similar phenomenon that the variance of stochastic estimators and the expected loss and error on both training and validation sets are decreasing functions of $b$ even if we train XLNet using Adam. This example gives us confidence that the decreasing properties are not merely restricted on shallow neural networks or vanilla SGD algorithms. They actually appear in many advanced models and optimization methods.

## B    LEMMAS AND PROOFS

### B.1    LEMMAS AND PROOFS OF RESULTS IN SECTION 3.1

For two matrices $A, B$ with the same dimension, we define the inner product $\langle A, B \rangle \triangleq \operatorname{tr}\left(A^T B\right)$.

**Lemma 3.** *Suppose that $f(x)$ and $g(x)$ are both smooth, non-negative and decreasing functions of $x \in \mathbb{R}$. Then $h(x) = f(x)g(x)$ is also a non-negative and decreasing function of $x$.*

*Proof.* It is obvious that $h(x)$ is non-negative for all $x$. The first-order derivative of $h$ is

$$h'(x) = f'(x)g(x) + f(x)g'(x) \leqslant 0,$$

and thus $h(x)$ is also a decreasing function of $x$. □

*Proof of Lemma 1.* Throughout the paper, We use $C_n^k = \frac{n!}{k!(n-k)!}$ to denote the combinatorial number. Note that

$$
\begin{aligned}
\mathbb{E}\left[g_t^b \left(g_t^b\right)^T \middle| \mathcal{F}_t^b\right] &= \frac{1}{b^2}\mathbb{E}\left[\sum_{i \in \mathcal{B}_t^b} \nabla L_i\left(w_t^b\right) \sum_{i \in \mathcal{B}_t^b} \nabla L_i\left(w_t^b\right)^T \middle| \mathcal{F}_t^b\right] \\
&= \frac{1}{b^2}\left(\frac{C_{n-1}^{b-1}}{C_n^b}\sum_{i=1}^n \nabla L_i\left(w_t^b\right)\nabla L_i\left(w_t^b\right)^T + \frac{C_{n-2}^{b-2}}{C_n^b}\sum_{i \neq j}\nabla L_i\left(w_t^b\right)\nabla L_j\left(w_t^b\right)^T\right) \\
&= \frac{1}{b^2}\left(\frac{b}{n}\sum_{i=1}^n \nabla L_i\left(w_t^b\right)\nabla L_i\left(w_t^b\right)^T + \frac{b(b-1)}{n(n-1)}\sum_{i \neq j}\nabla L_i\left(w_t^b\right)\nabla L_j\left(w_t^b\right)^T\right) \\
&= \frac{1}{b^2}\left(\frac{b(n-b)}{n(n-1)}\sum_{i=1}^n \nabla L_i\left(w_t^b\right)\nabla L_i\left(w_t^b\right)^T + \frac{b(b-1)}{n(n-1)}\sum_{i=1}^n\nabla L_i\left(w_t^b\right)\sum_{i=1}^n\nabla L_i\left(w_t^b\right)^T\right) \\
&= \frac{n-b}{bn(n-1)}\sum_{i=1}^n \nabla L_i\left(w_t^b\right)\nabla L_i\left(w_t^b\right)^T + \frac{(b-1)n}{b(n-1)}\nabla L\left(w_t^b\right)\nabla L\left(w_t^b\right)^T.
\end{aligned}
$$

For any $A \in \mathbb{R}^{p \times p}$, we have

$$
\begin{aligned}
\mathbb{E}\left[\left\|A g_t^b\right\|^2 \middle| \mathcal{F}_t^b\right] &= \mathbb{E}\left[\left(g_t^b\right)^T A^T A g_t^b \middle| \mathcal{F}_t^b\right] = \mathbb{E}\left[\operatorname{tr}\left(\left(g_t^b\right)^T A^T A g_t^b\right) \middle| \mathcal{F}_t^b\right] \\
&= \mathbb{E}\left[\operatorname{tr}\left(A^T A g_t^b \left(g_t^b\right)^T\right) \middle| \mathcal{F}_t^b\right] \\
&= \operatorname{tr}\left(A^T A \mathbb{E}\left[g_t^b \left(g_t^b\right)^T \middle| \mathcal{F}_t^b\right]\right) \\
&= \operatorname{tr}\left(\frac{n-b}{bn(n-1)} \sum_{i=1}^n A^T A \nabla L_i\left(w_t^b\right) \nabla L_i\left(w_t^b\right)^T + \frac{(b-1)n}{b(n-1)} A^T A \nabla L\left(w_t^b\right) \nabla L\left(w_t^b\right)^T\right) \\
&= \frac{n-b}{bn(n-1)} \sum_{i=1}^n \left\|A \nabla L_i\left(w_t^b\right)\right\|^2 + \frac{(b-1)n}{b(n-1)} \left\|A \nabla L\left(w_t^b\right)\right\|^2 \\
&= c_b \left(\frac{1}{n} \sum_{i=1}^n \left\|A \nabla L_i\left(w_t^b\right)\right\|^2 - \left\|A \nabla L\left(w_t^b\right)\right\|^2\right) + \left\|A \nabla L\left(w_t^b\right)\right\|^2 .
\end{aligned}
$$

Therefore, we have

$$
\begin{aligned}
\operatorname{var}\left(A g_t^b \middle| \mathcal{F}_t^b\right) &= \mathbb{E}\left[\left\|A g_t^b\right\|^2 \middle| \mathcal{F}_t^b\right] - \left\|\mathbb{E}\left[A g_t^b \middle| \mathcal{F}_t^b\right]\right\|^2 \\
&= \mathbb{E}\left[\left\|A g_t^b\right\|^2 \middle| \mathcal{F}_t^b\right] - \left\|A \nabla L\left(w_t^b\right)\right\|^2 \\
&= c_b \left(\frac{1}{n} \sum_{i=1}^n \left\|A \nabla L_i\left(w_t^b\right)\right\|^2 - \left\|A \nabla L\left(w_t^b\right)\right\|^2\right) .
\end{aligned}
$$

$\square$

**Lemma 4.** *For any set of square matrices $\{A_1, \cdots, A_n\} \in \mathbb{R}^{p \times p}$, if we denote $A = \sum_{i=1}^n A_i x_i x_i^T$, then we have*

$$
\mathbb{E}\left[\left\|\sum_{i=1}^n A_i \nabla L_i\left(w_{t+1}^b\right)\right\|^2 \middle| \mathcal{F}_0\right] = \mathbb{E}\left[\left\|\sum_{i=1}^n B_i \nabla L_i\left(w_t^b\right)\right\|^2 \middle| \mathcal{F}_0\right] + \frac{\alpha_t^2 c_b}{n^2} \sum_{k=1}^n \sum_{l=1}^n \mathbb{E}\left[\left\|\sum_{i=1}^n B_i^{kl} \nabla L_i\left(w_t^b\right)\right\|^2 \middle| \mathcal{F}_0\right].
$$

*Here $B_i = A_i - \frac{\alpha_t}{n} A$; $B_i^{kl} = A$ if $i = k, i \neq l$, $B_i^{kl} = A$ if $i = l, i \neq k$, and $B_i^{kl}$ equals the zero matrix, otherwise.*

*Proof of Lemma 4.* Let $C_i = x_i x_i^T$ and $C = \frac{1}{n} \sum_{i=1}^n C_i$. For the given $A_1, \ldots, A_n$, we denote $A = \sum_{i=1}^n A_i C_i$. Then we have

$$\mathbb{E}\left[ \left\| \sum_{i=1}^n A_i \nabla L_i\left(w_{t+1}^b\right) \right\|^2 \Big| \mathcal{F}_0 \right] = \mathbb{E}\left[ \mathbb{E}\left[ \left\| \sum_{i=1}^n A_i \nabla L_i\left(w_{t+1}^b\right) \right\|^2 \Big| \mathcal{F}_t^b \right] \Big| \mathcal{F}_0 \right]$$

$$= \mathbb{E}\left[ \mathbb{E}\left[ \left\| \sum_{i=1}^n A_i \left(x_i^T w_{t+1}^b - y_i\right) x_i \right\|^2 \Big| \mathcal{F}_t^b \right] \Big| \mathcal{F}_0 \right]$$

$$= \mathbb{E}\left[ \mathbb{E}\left[ \left\| \sum_{i=1}^n A_i \left(x_i^T \left(w_t^b - \alpha_t g_t^b\right) - y_i\right) x_i \right\|^2 \Big| \mathcal{F}_t^b \right] \Big| \mathcal{F}_0 \right]$$

$$= \mathbb{E}\left[ \mathbb{E}\left[ \left\| \sum_{i=1}^n A_i \nabla L_i\left(w_t^b\right) - \alpha_t A g_t^b \right\|^2 \Big| \mathcal{F}_t^b \right] \Big| \mathcal{F}_0 \right]$$

$$= \mathbb{E}\left[ \left\| \sum_{i=1}^n A_i \nabla L_i\left(w_t^b\right) \right\|^2 \Big| \mathcal{F}_0 \right] - 2\alpha_t \mathbb{E}\left[ \mathbb{E}\left[ \left\langle \sum_{i=1}^n A_i \nabla L_i\left(w_t^b\right), A g_t^b \right\rangle \Big| \mathcal{F}_t^b \right] \Big| \mathcal{F}_0 \right]$$

$$+ \alpha_t^2 \mathbb{E}\left[ \mathbb{E}\left[ \left\| A g_t^b \right\|^2 \Big| \mathcal{F}_t^b \right] \Big| \mathcal{F}_0 \right]$$

$$= \mathbb{E}\left[ \left\| \sum_{i=1}^n A_i \nabla L_i\left(w_t^b\right) \right\|^2 \Big| \mathcal{F}_0 \right] - 2\alpha_t \mathbb{E}\left[ \left\langle \sum_{i=1}^n A_i \nabla L_i\left(w_t^b\right), A \nabla L\left(w_t^b\right) \right\rangle \Big| \mathcal{F}_0 \right]$$

$$+ \alpha_t^2 \mathbb{E}\left[ c_b \left( \frac{1}{n} \sum_{i=1}^n \left\| A \nabla L_i(w_t^b) \right\|^2 - \left\| A \nabla L(w_t^b) \right\|^2 \right) + \left\| A \nabla L(w_t^b) \right\|^2 \Big| \mathcal{F}_0 \right]$$

$$= \mathbb{E}\left[ \left\| \sum_{i=1}^n A_i \nabla L_i\left(w_t^b\right) - \alpha_t A \nabla L(w_t^b) \right\|^2 \Big| \mathcal{F}_0 \right] + \alpha_t^2 c_b \mathbb{E}\left[ \frac{1}{n} \sum_{i=1}^n \left\| A \nabla L_i(w_t^b) \right\|^2 - \left\| A \nabla L(w_t^b) \right\|^2 \Big| \mathcal{F}_0 \right]$$

$$= \mathbb{E}\left[ \left\| \sum_{i=1}^n A_i \nabla L_i\left(w_t^b\right) - \alpha_t A \nabla L(w_t^b) \right\|^2 \Big| \mathcal{F}_0 \right] + \frac{\alpha_t^2 c_b}{n^2} \sum_{i \neq j} \mathbb{E}\left[ \left\| A \nabla L_i\left(w_t^b\right) - A \nabla L_j\left(w_t^b\right) \right\|^2 \Big| \mathcal{F}_0 \right]$$

$$= \mathbb{E}\left[ \left\| \sum_{i=1}^n \left(A_i - \frac{\alpha_t}{n} A\right) \nabla L_i\left(w_t^b\right) \right\|^2 \Big| \mathcal{F}_0 \right] + \frac{\alpha_t^2 c_b}{n^2} \sum_{i=1}^n \sum_{j=1}^n \mathbb{E}\left[ \left\| A \nabla L_i\left(w_t^b\right) - A \nabla L_j\left(w_t^b\right) \right\|^2 \Big| \mathcal{F}_0 \right].$$

Therefore, if we set $B_i = A_i - \frac{\alpha_t}{n} A$ and

$$B_i^{kl} = \begin{cases} A & i = k, i \neq l, \\ -A & i = l, i \neq k, \\ 0 & \text{otherwise}, \end{cases}$$

we have

$$\mathbb{E}\left[ \left\| \sum_{i=1}^n A_i \nabla L_i\left(w_{t+1}^b\right) \right\|^2 \Big| \mathcal{F}_0 \right] = \mathbb{E}\left[ \left\| \sum_{i=1}^n B_i \nabla L_i\left(w_t^b\right) \right\|^2 \Big| \mathcal{F}_0 \right] + \frac{\alpha_t^2 c_b}{n^2} \sum_{k=1}^n \sum_{l=1}^n \mathbb{E}\left[ \left\| \sum_{i=1}^n B_i^{kl} \nabla L_i\left(w_t^b\right) \right\|^2 \Big| \mathcal{F}_0 \right].$$

$\square$

*Proof of Theorem 1.* We use induction to show this statement.

When $t = 0$, $\mathbb{E}\left[ \left\| \sum_{i=1}^n A_i \nabla L_i\left(w_t^b\right) \right\|^2 \Big| \mathcal{F}_0 \right] = \left\| \sum_{i=1}^n A_i \nabla L_i\left(w_0\right) \right\|^2$ which is invariant of $b$. Therefore, it is a decreasing function of $b$.

Suppose the statement holds for $t$. For any set of matrices $\{A_1, \ldots, A_n\}$ in $\mathbb{R}^{p \times p}$, by Lemma 2 we know that there exist matrices $\{B_1, \cdots, B_n\}$ and $\{B_i^{kl} : i, k, l \in [n]\}$ such that

$$\mathbb{E}\left[\left\|\sum_{i=1}^n A_i \nabla L_i\left(w_{t+1}^b\right)\right\|^2 \middle| \mathcal{F}_0\right] = \mathbb{E}\left[\left\|\sum_{i=1}^n B_i \nabla L_i\left(w_t^b\right)\right\|^2 \middle| \mathcal{F}_0\right] + \frac{\alpha_t^2 c_b}{n^2} \sum_{k=1}^n \sum_{l=1}^n \mathbb{E}\left[\left\|\sum_{i=1}^n B_i^{kl} \nabla L_i\left(w_t^b\right)\right\|^2 \middle| \mathcal{F}_0\right].$$

By induction, we know that $\mathbb{E}\left[\left\|\sum_{i=1}^n B_i \nabla L_i\left(w_t^b\right)\right\|^2 \middle| \mathcal{F}_0\right]$ and all $\mathbb{E}\left[\left\|\sum_{i=1}^n B_i^{kl} \nabla L_i\left(w_t^b\right)\right\|^2 \middle| \mathcal{F}_0\right]$ are non-negative and decreasing functions of $b$. Besides, clearly $\frac{\alpha_t^2 c_b}{n^2} = \frac{\alpha_t^2(n-b)}{bn^3(n-1)}$ is a non-negative and decreasing function of $b$. By Lemma 3, we know that $\frac{\alpha_t^2 c_b}{n^2}\mathbb{E}\left[\left\|\sum_{i=1}^n B_i^{kl} \nabla L_i\left(w_t^b\right)\right\|^2 \middle| \mathcal{F}_0\right]$ is also a non-negative and decreasing function of $b$. Finally, $\mathbb{E}\left[\left\|\sum_{i=1}^n A_i \nabla L_i\left(w_{t+1}^b\right)\right\|^2 \middle| \mathcal{F}_0\right]$, as the sum of non-negative and decreasing functions in $b$, is a non-negative and decreasing function of $b$.

$\square$

In order to prove Theorem 2, we split the task to two separate theorems about the full gradient and the stochastic gradient and prove them one by one.

**Theorem 6.** *Fixing initial weights $w_0$, $\mathrm{var}\left(B\nabla L\left(w_t^b\right) \middle| \mathcal{F}_0\right)$ is a decreasing function of mini-batch size $b$ for all $b \in [n]$, $t \in \mathbb{N}$, and all square matrices $B \in \mathbb{R}^{p \times p}$.*

**Theorem 7.** *Fixing initial weights $w_0$, $\mathrm{var}\left(Bg_t^b \middle| \mathcal{F}_0\right)$ is a decreasing function of mini-batch size $b$ for all $b \in [n]$, $t \in \mathbb{N}$, and all square matrices $B \in \mathbb{R}^{p \times p}$.*

*Proof of Theorem 6.* We induct on $t$ to show that the statement holds. For $t = 0$, we have $\mathrm{var}\left(B\nabla L\left(w_t^b\right) \middle| \mathcal{F}_0\right) = 0$ for any matrix $B$. Suppose the statement holds for $t - 1 \geqslant 0$. Note that from

$$\begin{aligned}
\nabla L\left(w_t^b\right) &= \frac{1}{n}\sum_{i=1}^n x_i\left(x_i^T w_t^b - y_i\right) \\
&= \frac{1}{n}\sum_{i=1}^n x_i\left(x_i^T\left(w_{t-1}^b - \alpha_t g_{t-1}^b\right) - y_i\right) \\
&= \frac{1}{n}\sum_{i=1}^n x_i\left(x_i^T w_{t-1}^b - y_i\right) - \frac{\alpha_t}{n}\sum_{i=1}^n x_i x_i^T g_{t-1}^b \\
&= \nabla L\left(w_{t-1}^b\right) - \alpha_t C g_{t-1}^b,
\end{aligned}$$

we have

$$\text{var}\left(B\nabla L\left(w_t^b\right)\middle|\mathcal{F}_0\right)$$

$$= \text{var}\left(B\nabla L\left(w_{t-1}^b\right) - \alpha_t BCg_{t-1}^b\middle|\mathcal{F}_0\right)$$

$$= \mathbb{E}\left[\left\|B\nabla L\left(w_{t-1}^b\right) - \alpha_t BCg_{t-1}^b\right\|^2\middle|\mathcal{F}_0^b\right] - \left\|\mathbb{E}\left[B\nabla L\left(w_{t-1}^b\right) - \alpha_t BCg_{t-1}^b\middle|\mathcal{F}_0^b\right]\right\|^2$$

$$= \mathbb{E}\left[\left\|B\nabla L\left(w_{t-1}^b\right)\right\|^2 - 2\alpha_t\left\langle B\nabla L\left(w_{t-1}^b\right), BCg_{t-1}^b\right\rangle + \alpha_t^2\left\|BCg_{t-1}^b\right\|^2\middle|\mathcal{F}_0^b\right] - \left\|\mathbb{E}\left[B\nabla L\left(w_{t-1}^b\right) - \alpha_t BCg_{t-1}^b\middle|\mathcal{F}_0^b\right]\right\|^2$$

$$= \mathbb{E}\left[\left\|B\nabla L\left(w_{t-1}^b\right)\right\|^2\middle|\mathcal{F}_0\right] + \alpha_t^2\mathbb{E}\left[\mathbb{E}\left[\left\|BCg_{t-1}^b\right\|^2\middle|\mathcal{F}_{t-1}^b\right]\middle|\mathcal{F}_0^b\right] - 2\alpha_t\mathbb{E}\left[\mathbb{E}\left[\left\langle B\nabla L\left(w_{t-1}^b\right), BCg_{t-1}^b\right\rangle\middle|\mathcal{F}_{t-1}^b\right]\middle|\mathcal{F}_0\right]$$

$$\quad - \left\|\mathbb{E}\left[\mathbb{E}\left[B\nabla L\left(w_{t-1}^b\right) - \alpha_t BCg_{t-1}^b\middle|\mathcal{F}_{t-1}^b\right]\middle|\mathcal{F}_0^b\right]\right\|^2$$

$$= \mathbb{E}\left[\left\|B\nabla L\left(w_{t-1}^b\right)\right\|^2\middle|\mathcal{F}_0\right] + \alpha_t^2\mathbb{E}\left[c_b\left(\frac{1}{n}\sum_{i=1}^n\left\|BC\nabla L_i\left(w_{t-1}^b\right)\right\|^2 - \left\|BC\nabla L\left(w_{t-1}^b\right)\right\|^2\right) + \left\|BC\nabla L\left(w_{t-1}^b\right)\right\|^2\middle|\mathcal{F}_0\right]$$

$$\quad - 2\alpha_t\mathbb{E}\left[\left\langle B\nabla L\left(w_{t-1}^b\right), BC\nabla L\left(w_{t-1}^b\right)\right\rangle\middle|\mathcal{F}_0\right] - \left\|\mathbb{E}\left[B\nabla L\left(w_{t-1}^b\right) - \alpha_t BC\nabla L\left(w_{t-1}^b\right)\middle|\mathcal{F}_0^b\right]\right\|^2$$

$$\tag{3}$$

$$= \mathbb{E}\left[\left\|B\left(I - \alpha_t C\right)\nabla L\left(w_{t-1}^b\right)\right\|^2\middle|\mathcal{F}_0^b\right] + \alpha_t^2 c_b\mathbb{E}\left[\left(\frac{1}{n}\sum_{i=1}^n\left\|BC\nabla L_i\left(w_{t-1}^b\right)\right\|^2 - \left\|BC\nabla L\left(w_{t-1}^b\right)\right\|^2\right)\middle|\mathcal{F}_0\right]$$

$$\quad - \left\|\mathbb{E}\left[B\left(I - \alpha_t C\right)\nabla L\left(w_{t-1}^b\right)\middle|\mathcal{F}_0^b\right]\right\|^2$$

$$= \text{var}\left(B\left(I - \alpha_t C\right)\nabla L\left(w_{t-1}^b\right)\middle|\mathcal{F}_0\right) + \alpha_t^2 c_b\left(\frac{1}{n}\sum_{i=1}^n\mathbb{E}\left[\left\|BC\nabla L_i\left(w_{t-1}^b\right)\right\|^2\middle|\mathcal{F}_0\right] - \mathbb{E}\left[\left\|BC\nabla L\left(w_{t-1}^b\right)\right\|^2\middle|\mathcal{F}_0\right]\right)$$

$$= \text{var}\left(B\left(I - \alpha_t C\right)\nabla L\left(w_{t-1}^b\right)\middle|\mathcal{F}_0\right) + \frac{\alpha_t^2 c_b}{n^2}\sum_{i\neq j}\mathbb{E}\left[\left\|BC\nabla L_i\left(w_{t-1}^b\right) - BC\nabla L_j\left(w_{t-1}^b\right)\right\|^2\middle|\mathcal{F}_0\right],$$

$$\tag{4}$$

where (3) is by Lemma 1. By induction, we know that the first term of (4) is a decreasing function of $b$. Taking $A_i = BC, A_j = -BC, A_k = 0, k \in [n]\backslash\{i, j\}$ in Theorem 1, we know that

$$\mathbb{E}\left[\left\|BC\nabla L_i\left(w_{t-1}^b\right) - BC\nabla L_j\left(w_{t-1}^b\right)\right\|^2\middle|\mathcal{F}_0\right]$$

is also a decreasing function of $b$. Note that $\frac{\alpha_t^2 c_b}{n^2}$ decreases as $b$ increases. By Lemma 3 we learn that (4) is a decreasing function of $b$ and hence we have completed the induction.

$\square$

*Proof of Theorem 7.* We have

$$\text{var}\left(Bg_t^b\middle|\mathcal{F}_0\right) = \mathbb{E}\left[\left\|Bg_t^b\right\|^2\middle|\mathcal{F}_0\right] - \left\|\mathbb{E}\left[Bg_t^b\middle|\mathcal{F}_0\right]\right\|^2$$

$$= \mathbb{E}\left[\mathbb{E}\left[\left\|Bg_t^b\right\|^2\middle|\mathcal{F}_t^b\right]\middle|\mathcal{F}_0\right] - \left\|\mathbb{E}\left[\mathbb{E}\left[Bg_t^b\middle|\mathcal{F}_t^b\right]\middle|\mathcal{F}_0\right]\right\|^2$$

$$= c_b\left(\frac{1}{n}\sum_{i=1}^n\mathbb{E}\left[\left\|B\nabla L_i\left(w_t^b\right)\right\|^2\middle|\mathcal{F}_0\right] - \mathbb{E}\left[\left\|B\nabla L\left(w_t^b\right)\right\|^2\middle|\mathcal{F}_0\right]\right)$$

$$\quad + \mathbb{E}\left[\left\|B\nabla L\left(w_t^b\right)\right\|^2\middle|\mathcal{F}_0\right] - \left\|\mathbb{E}\left[B\nabla L\left(w_t^b\right)\middle|\mathcal{F}_0\right]\right\|^2$$

$$= \frac{c_b}{n^2}\sum_{i\neq j}\mathbb{E}\left[\left\|B\nabla L_i\left(w_t^b\right) - B\nabla L_j\left(w_t^b\right)\right\|^2\middle|\mathcal{F}_0\right] + \text{var}\left(B\nabla L\left(w_t^b\right)\middle|\mathcal{F}_0\right).$$

Taking $A_i = B, A_j = -B, A_k = 0, k \in [n]\backslash\{i, j\}$ in Theorem 1, we know that

$$\mathbb{E}\left[\left\|B\nabla L_i\left(w_t^b\right) - B\nabla L_j\left(w_t^b\right)\right\|^2\middle|\mathcal{F}_0\right]$$

is a decreasing and non-negative function of $b$ for all $i, j \in [n]$. By Theorem 6, we know that $\text{var}\left(B\nabla L\left(w_t^b\right)\middle|\mathcal{F}_0\right)$ is also a decreasing function of $b$. Therefore, $\text{var}\left(Bg_t^b\middle|\mathcal{F}_0\right)$, as the sum of two decreasing functions of $b$, is also a decreasing function of $b$. $\square$

*Proof of Corollary 1.* Simply taking $B = I_p$ in Theorem 1 yields the proof. $\qquad\square$

## B.2 PROOFS FOR RESULTS IN 3.2

**Remark.** We often rely on the trivial facts that $x_1 x_2^T = x_1 I_p x_2^T$ and $x_1 x_2^T x_3 x_4^T = x_1 x_2^T I_p x_3 x_4^T$.

**Lemma 5.** *Given a multiplicative term of parameter matrices $\{u_i v_i^T : u_i, v_i \in \mathbb{R}^p, i \in [n_1]\} \cup \{A_j : A_j \in \mathbb{R}^{p \times p}, j \in [n_2]\}$ and constant matrix $\{I_p\}$ such that $\deg(u_1 v_1^T; M) \geqslant 1$, we have*

$$\mathrm{tr}\,(M) = v_1^T M' u_1,$$

*where $M'$ is a multiplicative term of parameter matrices $\{u_i v_i^T : u_i, v_i \in \mathbb{R}^p, i \in [n_1]\} \cup \{A_j : A_j \in \mathbb{R}^{p \times p}, j \in [n_2]\}$ and constant matrix $\{I_p\}$ such that $\deg(M) = \deg(M') + 1, \deg(A_j; M) = \deg(A_j; M'), j \in [n_2], \deg(u_i v_i^T; M) = \deg(u_i v_i^T; M'), i \in [2 : n_1]$ and $\deg(u_1 v_1^T; M) = \deg(u_1 v_1^T; M') + 1$.*

*Proof.* By the definition of multiplicative terms, we know that there exist two multiplicative terms $M_1, M_2$ of parameter matrices $\{u_i v_i^T : u_i, v_i \in \mathbb{R}^p, i \in [n_1]\} \cup \{A_j : A_j \in \mathbb{R}^{p \times p}, j \in [n_2]\}$ and constant matrix $\{I_p\}$ such that

$$M = M_1 u_1 v_1^T M_2,$$

where $\deg(M) = \deg(M_1) + \deg(M_2) + 1, \deg(A_j; M) = \deg(A_j; M_1) + \deg(A_j; M_2), j \in [n_2], \deg(u_i v_i^T; M) = \deg(u_i v_i^T; M_1) + \deg(u_i v_i^T; M_2), i \in [2 : n_1]$ and $\deg(u_1 v_1^T; M) = \deg(u_1 v_1^T; M_1) + \deg(u_1 v_1^T; M_2) + 1$. Therefore we have

$$\mathrm{tr}\,(M) = \mathrm{tr}\,\left(M_1 u_1 v_1^T M_2\right) = \mathrm{tr}\,\left(v_1^T M_2 M_1 u_1\right) = v_1^T M_2 M_1 u_1.$$

Note that $M' = M_2 M_1$ satisfies that $\deg(M') = \deg(M_1) + \deg(M_2), \deg(A_j, M') = \deg(A_j; M_1) + \deg(A_j; M_2), j \in [n_2], \deg(u_i v_i^T; M) = \deg(u_i v_i^T; M_1) + \deg(u_i v_i^T; M_2), i \in [2 : n_1]$ and $\deg(u_1 v_1^T; M') = \deg(u_1 v_1^T; M_1) + \deg(u_1 v_1^T; M_2) + 1$. We have finished the proof. $\qquad\square$

The following two lemmas focus on the expectation of the product of quadratic forms of the standard normal samples. Lemma 6 focuses on single sample while 7 focuses on the same form with $b$ i.i.d. samples drawn from the standard normal distribution.

**Lemma 6.** *Given matrices $A_j \in \mathbb{R}^{p \times p}, j \in [m - 1]$, we have*

$$\mathbb{E}_{x \sim \mathcal{N}(0, I_p)}\left[xx^T A_1 xx^T A_2 \cdots A_{m-1} xx^T\right] = \sum_{i=1}^{N_m} \prod_{k=1}^{n_i} \mathrm{tr}\,(M_{ik})\, M_{i0},$$

*where $N_m$ and $n_i, i \in [N_m]$ are constants depending on $m$ and $\{M_{ik}, k \in [0 : n_i], i \in [N_m]\}$ are multiplicative terms of parameter matrices $\{A_j, j \in [m-1]\}$ and constant matrix $\{I_p\}$. Furthermore, for every $i \in [N_m]$, we have $\sum_{k=0}^{n_i} \deg(A_j; M_{ik}) = 1, j \in [m-1]$ and therefore $\sum_{k=0}^{n_i} \deg(M_{ik}) = m - 1$.*

*Proof.* See Magnus (1978). $\qquad\square$

**Lemma 7.** *We are given matrices $A_j \in \mathbb{R}^{p \times p}, j \in [m-1]$ and random vectors $x_i, i \in [b]$ independently and identically drawn from $\mathcal{N}(0, I_p)$. We assume that the multi-set $\mathcal{S} = \{i_j, i_j' : j \in [m]\}$ satisfies that for every $i \in \mathcal{S}$, $i$ is an element of $[b]$ and the number of appearance of $i$ in $\mathcal{S}$ is even. Then*

$$\mathbb{E}_{x_i \sim \mathcal{N}(0, I_p)}\left[x_{i_1} x_{i_1'}^T A_1 x_{i_2} x_{i_2'}^T A_2 \cdots A_{m-1} x_{i_m} x_{i_m'}^T\right] = \sum_{i=1}^{N_m} \prod_{k=1}^{n_i} \mathrm{tr}\,(M_{ik})\, M_{i0}, \qquad (5)$$

*where $N_m$ and $n_i$ are constants depending on $m$ (and independent of $b$) and $M_{ik}, k \in [0 : n_i], i \in [N_m]$ are multiplicative terms of parameter matrices $\{A_j, j \in [m-1]\}$ and constant matrix $\{I_p\}$. Furthermore, for every $i \in [N_m]$, we have $\sum_{k=0}^{n_i} \deg(A_j; M_{ik}) = 1, j \in [m-1]$ and therefore $\sum_{k=0}^{n_i} \deg(M_{ik}) = m - 1$.*

*Proof.* Let $\beta_i, i \in [b]$ be the number of appearances of $i$ in $\mathcal{S}$, which are even by assumption. We induct on the quantity $N = \sum_{i=1}^{b} \mathbb{1}\{\beta_i \neq 0\}$.

For the base case of $N = 1$, all elements in the multi-set $\mathcal{S}$ have the same value. Without loss of generality, we assume $i_j = i'_j = 1, j \in [m]$. Then

$$\mathbb{E}_{x_i \sim \mathcal{N}(0, I_p)} \left[ x_{i_1} x_{i'_1}^T A_1 x_{i_2} x_{i'_2}^T \cdots A_{m-1} x_{i_m} x_{i'_m}^T \right] = \mathbb{E}_{x_1 \sim \mathcal{N}(0, I_p)} \left[ x_1 x_1^T A_1 x_1 x_1^T \cdots A_{m-1} x_1 x_1^T \right],$$

which is the statement of Lemma 6.

Suppose the statement holds for $N \geqslant 1$, and we consider the case of $N + 1$. Note that $x_{i'_j}^T A_j x_{i_{j+1}} = x_{i_{j+1}}^T A_j x_{i'_j}$ is a scalar so that we can move it around without changing the value of the expression[2]. We distinguish two cases.

- Let $i_1 \neq i'_m$. Without loss of generality, we assume $i_1 = 1$. We can always change the order of $x_{i'_j}^T A_j x_{i_{j+1}}, j \in [m-1]$ (and flip it to be $x_{i_{j+1}}^T A_j x_{i'_j}$ if necessary) such that all $x_1$'s appear in the form of $x_1 x_1^T$:

$$x_{i_1} x_{i'_1}^T A_1 x_{i_2} x_{i'_2}^T A_2 \cdots A_{m-1} x_{i_m} x_{i'_m}^T = x_1 \left( x_{i'_1}^T A_1 x_{i_2} x_{i'_2}^T A_2 \cdots A_{m-1} x_{i_m} \right) x_{i'_m}^T$$
$$= x_1 x_1^T \widetilde{A}_1 x_1 x_1^T \widetilde{A}_2 \cdots \widetilde{A}_{\frac{\beta_1}{2}-1} x_1 x_1^T \widetilde{A}_{\frac{\beta_1}{2}} \widetilde{x} x_{i'_m}^T$$

where $\widetilde{x} \in \{x_i, i \in [b]\}, \widetilde{x} \neq x_1$ and $\widetilde{A}_i$'s are multiplicative terms of parameter matrices $\{x_u x_v^T : u, v \in [2:b]\} \cup \{A_j : j \in [m-1]\}$ and constant matrix $\{I_p\}$ such that $\sum_{u,v \in [2:b]} \sum_{k=1}^{\frac{\beta_1}{2}} \deg(x_u x_v^T; \widetilde{A}_k) = m - \frac{\beta_1}{2} - 1$ and $\sum_{k=1}^{\frac{\beta_1}{2}} \deg(A_j; \widetilde{A}_k) = 1, j \in [m-1]$[3].

Applying Lemma 6 and the law of iterative expectations, we have

$$\mathbb{E}_{x_i \sim \mathcal{N}(0, I_p)} \left[ x_{i_1} x_{i'_1}^T A_1 x_{i_2} x_{i'_2}^T \cdots A_{m-1} x_{i_m} x_{i'_m}^T \right] = \mathbb{E}_{x_1, \cdots, x_b} \left[ x_1 x_1^T \widetilde{A}_1 x_1 x_1^T \widetilde{A}_2 \cdots \widetilde{A}_{\frac{\beta_1}{2}-1} x_1 x_1^T \widetilde{A}_{\frac{\beta_1}{2}} \widetilde{x} x_{i'_m}^T \right]$$
$$= \mathbb{E}_{x_2, \cdots, x_b} \left[ \left( \sum_{i=1}^{N_m} \prod_{k=1}^{n_i} \operatorname{tr}(M_{ik}) M_{i0} \right) \widetilde{A}_{\frac{\beta_1}{2}} \widetilde{x} x_{i'_m}^T \right]$$
$$= \sum_{i=1}^{N_m} \mathbb{E}_{x_2, \cdots, x_b} \left[ \left( \prod_{k=1}^{n_i} \operatorname{tr}(M_{ik}) M_{i0} \right) \widetilde{A}_{\frac{\beta_1}{2}} \widetilde{x} x_{i'_m}^T \right],$$

where $N_m$ and $n_i$ are constant depending on $m$ (and independent of $b$) and $M_{ik}, k \in [0:n_i], i \in [N_m]$ are multiplicative terms of parameter matrices $\left\{ \widetilde{A}_j, j \in [\frac{\beta_1}{2} - 1] \right\}$ and constant matrix $\{I_p\}$. Furthermore, for every $i \in [N_m]$, we have $\sum_{k=0}^{n_i} \deg(\widetilde{A}_j; M_{ik}) = 1, j \in [\frac{\beta_1}{2} - 1]$ and therefore $\sum_{k=0}^{n_i} \deg(M_{ik}) = \frac{\beta_1}{2} - 1$.

Combining the definition of $\widetilde{A}_j$'s, we know that $M_{ik}, k \in [0:n_i], i \in [N_m]$ are multiplicative terms of parameter matrices $\{x_u x_v^T : u, v \in [2:b]\} \cup \{A_j : j \in [m-1]\}$ and constant

---

[2]For example, we can rewrite

$$x_{i_1} x_{i'_1}^T A_1 x_{i_2} x_{i'_2}^T A_2 x_{i_3} x_{i'_3}^T = x_{i_1} \left( x_{i'_1}^T A_1 x_{i_2} \right) \left[ x_{i'_2}^T A_2 x_{i_3} \right] x_{i'_3}^T = x_{i_1} \left[ x_{i'_2}^T A_2 x_{i_3} \right] \left( x_{i'_1}^T A_1 x_{i_2} \right) x_{i'_3}^T$$
$$= x_{i_1} \left[ x_{i'_2}^T \left( x_{i'_1}^T A_1 x_{i_2} \right) A_2 x_{i_3} \right] x_{i'_3}^T = x_{i_1} \left[ x_{i'_2}^T A_2 \left( x_{i'_1}^T A_1 x_{i_2} \right) x_{i_3} \right] x_{i'_3}^T.$$

[3]For example, we can rewrite

$$x_1 x_2^T A_1 x_1 x_1^T A_2 x_3 x_3^T A_3 x_1 x_2 = x_1 \left( x_2^T A_1 x_1 \right) \left[ x_1^T A_2 x_3 \right] \left\{ x_3^T A_3 x_1 \right\} x_2 = x_1 \left( x_1^T A_1 x_2 \right) \left[ x_3^T A_2 x_1 \right] \left\{ x_1^T A_3 x_3 \right\} x_2$$
$$= x_1 x_1^T A_1 x_2 x_3^T A_2 x_1 x_1^T A_3 x_3 x_2 = x_1 x_1^T \widetilde{A}_1 x_1 x_1^T \widetilde{A}_2 \widetilde{x} x_2,$$

where $\widetilde{A}_1 = A_1 x_2 x_3^T A_2, \widetilde{A}_2 = A_3$ and $\widetilde{x} = x_3$. Besides, $m = 4, \beta_1 = 4$, thus the degree of $x_u x_v^T$ in all $\widetilde{A}_k$ sum up to $m - \frac{\beta_1}{2} - 1 = 1$

matrix $\{I_p\}$ such that for every $i \in [N_m]$, we have $\sum_{u,v \in [2:b]} \sum_{k=0}^{n_i} \deg(x_u x_v^T; M_{ik}) = m - \frac{\beta_1}{2} - 1$ and $\sum_{k=0}^{n_i} \deg(A_j; M_{ik}) = 1, j \in [m-1]$.

Applying Lemma 5, for every $k \in [0:n_i]$ and every $i \in [N_m]$, there exists $u_{ik}, v_{ik} \in \{x_j : j \in [2:b]\}$ and multiplicative term $M'_{ik}$ of parameter matrices $\{x_u x_v^T : u, v \in [2:b]\} \cup \{A_j : j \in [m-1]\}$ and constant matrix $\{I_p\}$ such that

$$\mathrm{tr}\,(M_{ik}) = u_{ik}^T M'_{ik} v_{ik}.$$

Therefore, we have

$$\left( \prod_{k=1}^{n_i} \mathrm{tr}\,(M_{ik})\, M_{i0} \right) \widetilde{A}_{\frac{\beta_1}{2}} \widetilde{x} x_{i'_m}^T = \prod_{k=1}^{n_i} \left( u_{ik}^T M'_{ik} v_{ik} \right) M_{i0} \widetilde{A}_{\frac{\beta_1}{2}} \widetilde{x} x_{i'_m}^T = M_{i0} \widetilde{A}_{\frac{\beta_1}{2}} \widetilde{x} \prod_{k=1}^{n_i} \left( u_{ik}^T M'_{ik} v_{ik} \right) x_{i'_m}^T \triangleq U_i.$$

Note that for every $i \in [N_m]$, we have

$$\sum_{j=1}^{m-1} \deg(x_i; A_j) = \sum_{k=1}^{n_i} \deg(x_i; M'_{ik}) + \deg(x_i; M_{i0}) + \deg \left( x_i; \widetilde{A}_{\frac{\beta_1}{2}} \right) + \deg(x_i; \widetilde{x}) + \deg \left( x_i; x_{i'_m}^T \right),$$

and for every $j \in [m-1]$, we have

$$\sum_{k=1}^{n_i} \deg(A_j; M'_{ik}) + \deg(A_j; M_{i0}) + \deg \left( A_j; \widetilde{A}_{\frac{\beta_1}{2}} \right) = 1.$$

In other words, for every $i \in [N_m]$, $U_i$ has the form of $\widehat{A}_0 x_{\widehat{i}_1} x_{\widehat{i}'_1}^T \widehat{A}_1 x_{\widehat{i}_2} x_{\widehat{i}'_2}^T \cdots \widehat{A}_{m-1} x_{\widehat{i}_{m'}} x_{i'_m}^T \widehat{A}_{m'}$ but there is no appearance of $x_1$. Here $x_{\widehat{i}_j}, x_{\widehat{i}_j} \in \{x_j, j \in [2:b]\}$, and $\widehat{A}_i, i \in [0:m]$ are multiplicative terms of parameter matrices $\{A_j, j \in [m-1]\}$ and constant matrix $\{I_p\}$. Furthermore, for every $j \in [m-1]$, we have $\sum_{k=0}^{n_i} \deg(A_j; \widehat{A}_i) = 1$. Note that here we use the liberty of adding identity matrices if more than two consecutive $x$'s appear. Since we have reduced $N+1$ by one, we can use induction on $x_{\widehat{i}_1} x_{\widehat{i}'_1}^T \widehat{A}_1 x_{\widehat{i}_2} x_{\widehat{i}'_2}^T \cdots \widehat{A}_{m-1} x_{\widehat{i}_{m'}} x_{i'_m}^T$ and finish the proof. The two constant matrices $\widehat{A}_0$ and $\widehat{A}_m$ do not change the result of expectation since $\mathbb{E} \left( \widehat{A}_0 X \widehat{A}_{m'} \right) = \widehat{A}_0 \mathbb{E}(X) \widehat{A}_{m'}$.

- If $i_1 = i'_m$, without loss of generality we assume, $i'_1 = 1$ and $i'_1 \neq i_1$ (note that all $x_{i'_j}^T A_j x_{i_{j+1}}, j \in [m-1]$ are inter-changeable and there is at least one element in $\mathcal{S}$ that is not equal to $i_1$). We change the orders of $x_{i'_j}^T A_j x_{i_{j+1}}, j \in [m-1]$ (and flip it to be $x_{i_{j+1}}^T A_j x_{i'_j}$ if necessary) such that all $x_1$'s appear in a consecutive form of $x_1 x_1^T$:

$$x_{i_1} x_{i'_1}^T A_1 x_{i_2} x_{i'_2}^T A_2 \cdots A_{m-1} x_{i_m} x_{i'_m}^T = x_{i_1} \left( x_{i'_1}^T A_1 x_{i_2} x_{i'_2}^T A_2 \cdots A_{m-1} x_{i_m} \right) x_{i'_m}^T$$
$$= x_{i_1} \left( \widetilde{x}_1^T \widetilde{A}_0 \left[ x_1 x_1^T \widetilde{A}_1 \cdots \widetilde{A}_{\frac{\beta_1}{2}-1} x_1 x_1^T \right] \widetilde{A}_{\frac{\beta_1}{2}} \widetilde{x}_2 \right) x_{i'_m}^T,$$

where $\widetilde{x}_1, \widetilde{x}_2 \in \{x_i, i \in [b]\}, \widetilde{x}_1, \widetilde{x}_2 \neq x_1$ and $\widetilde{A}_i$'s are multiplicative terms of parameter matrices $\{x_u x_v^T : u, v \in [2:b]\} \cup \{A_j : j \in [m-1]\}$ and constant matrix $\{I_p\}$ such that

$$\sum_{u,v \in [2:b]} \sum_{k=0}^{\frac{\beta_1}{2}} \deg(x_u x_v^T; \widetilde{A}_k) = m - \frac{\beta_1}{2} - 2$$

and $\sum_{k=0}^{\frac{\beta_1}{2}} \deg(A_j; \widetilde{A}_k) = 1, j \in [m-1]$. The remaining reasoning is the same as the previous case.

$\square$

**Remark.** If one of the $\beta_i$ numbers of appearance of $x_j, j \in [b]$ is odd, then it is easy to see that the result in (5) is the zero matrix.

As pointed out in the Section 1, the difficulty of studying the dynamics of SGD is how to connect the quantities in iteration $t$ with fixed variables, like initial weights $W_{0,1}, W_{0,2}$ and mini-batch size $b$. We overcome this challenge by the following two lemmas. Lemma 8 provides the relationship between $g_{t,i}^b, i = 1, 2$ and $W_{t,i}^b, i = 1, 2$ by taking expectation over the distribution of random samples in $\mathcal{B}_t^b$. Lemma 9 shows the relationship between $W_{t,i}^b, i = 1, 2$ and $g_{t-1,i}^b, i = 1, 2$ using (1) and (2).

**Lemma 8.** *For multiplicative terms $M_i, i \in [0:m]$ of parameter matrices $\{g_{t,1}^b, g_{t,2}^b\}$ and constant matrices $\{W_{t,1}^b, W_{t,2}^b, W_1^*, W_2^*\}$ with degree $d_i$, respectively, we denote $M = \prod_{i=1}^m \mathrm{tr}(M_i) M_0$ and $d = \sum_{i=0}^m d_i$. There exists a set of multiplicative terms $\{M_{ij}^k, i \in [m_k], j \in [0:m_{ki}], k \in [0:q]\}$ of parameter matrices $\{W_{t,1}^b, W_{t,2}^b\}$ and constant matrices $\{W_1^*, W_2^*\}$ such that*

$$\mathbb{E}\left[M \big| \mathcal{F}_t^b\right] = N_0 + N_1 \frac{1}{b} + \cdots + N_d \frac{1}{b^d},$$

*where $N_k = \sum_{i=1}^{m_k} \prod_{j=1}^{m_{ki}} \mathrm{tr}\left(M_{ij}^k\right) M_{i0}^k, k \in [0:d]$. Here $m_k, m_{ki}$ are constants independent of $b$, and $\sum_{j=0}^{m_{ki}} \deg\left(M_{ij}^k\right) \leqslant 3d + \sum_{i=0}^m \left(\deg\left(W_{t,1}^b; M_i\right) + \deg(W_{t,2}^b; M_i)\right)$.*

**Lemma 9.** *For multiplicative term $M_i, i \in [0:m]$ of parameter matrices $\{W_{t,1}^b, W_{t,2}^b\}$ and constant matrices $\{W_1^*, W_2^*\}$ of degree $d_i$, let $d = 2^{d_0 + \cdots + d_m}$. There exists a set of multiplicative terms $\{M_{ik}, i \in [0:m], k \in [d]\}$ of parameter matrices $\{g_{t,1}^b, g_{t,2}^b\}$ and constant matrices $\{W_{t,1}^b, W_{t,2}^b, W_1^*, W_2^*\}$ such that*

$$\prod_{i=1}^m \mathrm{tr}(M_i) M_0 = \sum_{k=1}^d \prod_{i=1}^m \mathrm{tr}(M_{ik}) M_{0k},$$

*where $\sum_{i=0}^m \deg(M_{ik}) \leqslant d$.*

*Proof of Lemma 8.* By (1) and (2) we have

$$M = \prod_{i=1}^m \mathrm{tr}(M_i) M_0 = \frac{1}{b^d} \sum_{k=1}^{b^d} \prod_{i=1}^m \mathrm{tr}(M_{ki}) M_{k0}, \tag{6}$$

where each $M_{ki}, k \in [b^d], i \in [0:m]$ is a multiplicative term of parameter matrices $\{x_{t,i} x_{t,i}^T, i \in [b]\}$ and constant matrices $\{W_{t,1}^b, W_{t,2}^b, \mathcal{W}_t^b\}$. Let $\widetilde{M}_k = \prod_{i=1}^m \mathrm{tr}(M_{ki}) M_{k0}, k \in [b^d]$. We split set $\left\{\widetilde{M}_k : k \in [b^d]\right\}$ into disjoint and non-empty sets (equivalent classes) $S_1, \ldots, S_{n_M}$ such that

1. for every $i \in [n_M]$ and every $M_1, M_2 \in S_i$, we have $\mathbb{E}\left[M_1 \big| \mathcal{F}_t^b\right] = \mathbb{E}\left[M_2 \big| \mathcal{F}_t^b\right]$,

2. for every $i, j \in [n_M], i \neq j$ and every $M_1 \in S_i$ and $M_2 \in S_j$, we have $\mathbb{E}\left[M_1 \big| \mathcal{F}_t^b\right] \neq \mathbb{E}\left[M_2 \big| \mathcal{F}_t^b\right]$.

Note that $\cup_{i=1}^{n_M} S_i = \left\{\widetilde{M}_k : k \in [b^d]\right\}$. Let $\widehat{M}_k \in S_k$ represent the equivalent class $S_k$ (it can be any member of $S_k$). For every $i \in [n_M]$, we can always write $|S_i| = e_{i,0} + e_{i,1} b + \cdots + e_{i,d} b^d$ such that $e_{i,j} \in \mathbb{N}, e_{i,j} < b, j \in [0:d]$ (actually $e_{i,j}$'s are the digits of the base-$b$ representation of $|S_i|$). Then

we have

$$\mathbb{E}\left[M|\mathcal{F}_t^b\right] = \mathbb{E}\left[\frac{1}{b^d}\sum_{k=1}^{b^d}\widetilde{M}_k\middle|\mathcal{F}_t^b\right] = \frac{1}{b^d}\mathbb{E}\left[\sum_{i=1}^{n_M}\left(e_{i,0}+e_{i,1}b+\cdots+e_{i,d}b^d\right)\widehat{M}_i\middle|\mathcal{F}_t^b\right]$$

$$= \frac{1}{b^d}\sum_{i=1}^{n_M}\left(e_{i,0}+e_{i,1}b+\cdots+e_{i,d}b^d\right)\mathbb{E}\left[\widehat{M}_i\middle|\mathcal{F}_t^b\right] \tag{7}$$

$$= \sum_{i=1}^{n_M}\left(e_{i,d}+e_{i,d-1}\frac{1}{b}+\cdots+e_{i,0}\frac{1}{b^d}\right)\mathbb{E}\left[\widehat{M}_i\middle|\mathcal{F}_t^b\right].$$

It is important to note that $n_M$, the number of different equivalent classes, is independent of $b$. This follows from the fact that each $\mathbb{E}\left[\widetilde{M}_k\middle|\mathcal{F}_t^b\right]$ (and so as $\mathbb{E}\left[\widehat{M}_k\middle|\mathcal{F}_t^b\right]$) includes a finite number of weight matrices $W_{t,1}^b$ and $W_{t,2}^b$ with degree less than or equal to $3d + \sum_{i=0}^m\left(\deg\left(W_{t,1}^b;M_i\right)+\deg(W_{t,2}^b;M_i)\right)$ (see Lemma 7). Thus the number of partition sets is bounded by a quantity independent of $b$.

Note that each $M_{ki}$ can be represented as

$$M_{ki} = A_0^{ki}x_{t,i_1}^{ki}x_{t,i_1}^{ki}{}^T A_1^{ki}\cdots A_{d_i-1}^{ki}x_{t,i_{d_i}}^{ki}x_{t,i_{d_i}}^{ki}{}^T A_{d_i}^{ki}$$

for some matrices $A_0^{ki},\ldots,A_{d_i}^{ki}$ that are multiplicative term of parameter matrices $\{W_{t,1}^b, W_{t,2}^b and \mathcal{W}_t^b\}$ constant matrix $\{I_p\}$ (we stress again that some $A$ matrices can be identities, based on the definition of multiplicative terms), and $x_{t,i_1}^{ki},\ldots,x_{t,i_{d_i}}^{ki} \in \{x_{t,1},\ldots,x_{t,b}\}$. We have

$$\text{tr}\left(M_{ki}\right) = \text{tr}\left(A_0^{ki}x_{t,i_1}^{ki}x_{t,i_1}^{ki}{}^T A_1^{ki}\cdots A_{d_i-1}^{ki}x_{t,i_{d_i}}^{ki}x_{t,i_{d_i}}^{ki}{}^T A_{d_i}^{ki}\right)$$

$$= x_{t,i_{d_i}}^{ki}{}^T A_{d_i}^{ki}A_0^{ki}x_{t,i_1}^{ki}x_{t,i_1}^{ki}{}^T A_1^{ki}\cdots A_{d_i-1}^{ki}x_{t,i_{d_i}}^{ki}.$$

For every $k \in \left[b^d\right]$, we have

$$\prod_{i=1}^m\text{tr}\left(M_{ki}\right)M_{k0} = \left[\prod_{i=1}^m x_{t,i_{d_i}}^{ki}{}^T A_{d_i}^{ki}A_0^{ki}x_{t,i_1}^{ki}x_{t,i_1}^{ki}{}^T A_1^{ki}\cdots A_{d_i-1}^{ki}x_{t,i_{d_i}}^{ki}\right]A_0^{k0}x_{t,i_1}^{k0}x_{t,i_1}^{k0}{}^T A_1^{k0}\cdots A_{d_0-1}^{k0}x_{t,i_{d_0}}^{k0}x_{t,i_{d_0}}^{k0}{}^T A_{d_0}^{k0}$$

$$= \left[\prod_{i=1}^m x_{t,i_{d_i}}^{ki}{}^T A_{d_i}^{ki}A_0^{ki}x_{t,i_1}^{ki}x_{t,i_1}^{ki}{}^T A_1^{ki}\cdots A_{d_i-1}^{ki}x_{t,i_{d_i}}^{ki}\right]\left[x_{t,i_1}^{k0}{}^T A_1^{k0}\cdots A_{d_0-1}^{k0}x_{t,i_{d_0}}^{k0}\right]A_0^{k0}x_{t,i_1}^{k0}x_{t,i_{d_0}}^{k0}{}^T A_{d_0}^{k0},$$

which can be rewritten as

$$\widetilde{M}_k = \prod_{i=1}^m\text{tr}\left(M_{ki}\right)M_{k0} = \left(\prod_{j=1}^d x_{t,\bar{i}_j}^T A_j^k x_{t,\bar{i}'_j}\right)A_0^{k0}x_{t,i_1}^{k0}x_{t,i_{d_0}}^{k0}{}^T A_{d_0}^{k0}.$$

Note that the randomness of each $\widetilde{M}_k$ given $\mathcal{F}_t^b$ only comes from the randomness of $x_{t,j}$'s, i.e. for all $k \in \left[b^d\right]$ we have

$$\mathbb{E}\left[\widetilde{M}_k\middle|\mathcal{F}_t^b\right] = \mathbb{E}_{x_{t,j}\sim\mathcal{N}(0,I)}\left[\left(\prod_{j=1}^d x_{t,i_j}^T A_j^k x_{t,i'_j}\right)A_0^k x_{t,i'_0}x_{t,i_0}^T A_0^{k'}\right]$$

$$= \mathbb{E}_{x_{t,j}\sim\mathcal{N}(0,I)}\left[A_0^k x_{t,i'_0}\left(\prod_{j=1}^d x_{t,i_j}^T A_j^k x_{t,i'_j}\right)x_{t,i_0}^T A_0^{k'}\right] \tag{8}$$

$$= \sum_{i=1}^{n_M^k}\prod_{j=1}^{n_i^k}\text{tr}\left(\widetilde{M}_{ij}^k\right)\widetilde{M}_{i0}^k,$$

where the last equation comes from Lemma 7. Here $n_M^k, n_i^k, i \in \left[n_M^k\right], k \in \left[b^d\right]$ are constants independent of $b$, $M_{ij}^k$'s are multiplicative terms of parameter matrices $\left\{W_{t,1}^b, W_{t,2}^b, \mathcal{W}_t^b\right\}$ and constant matrix $\{I_p\}$ such that for every $i \in \left[n_M^k\right]$, we have

$$\sum_{j=0}^{n_i^k} \deg\left(\mathcal{W}_t^b; \widetilde{M}_{ij}^k\right) = d \tag{9}$$

and

$$\sum_{j=0}^{n_i^k} \left(\deg\left(W_{t,1}^b; \widetilde{M}_{ij}^k\right) + \deg\left(W_{t,2}^b; \widetilde{M}_{ij}^k\right)\right) = d + \sum_{r=0}^{m} \left(\deg\left(W_{t,1}^b; M_r\right) + \deg(W_{t,2}^b; M_r)\right). \tag{10}$$

These degree relationships can be observed from (1), (2), and the fact that each $g_{t,1}^b$ or $g_{t,1}^b$ contributes one $\mathcal{W}_t^b$ and one of $W_{t,1}^b$ or $W_{t,2}^b$ in $\prod_{j=1}^{n_i^k} \mathrm{tr}\left(\widetilde{M}_{ij}^k\right) \widetilde{M}_{i0}^k$. Note that $\mathcal{W}_t = W_{t,2}^b W_{t,2}^b - W_2^* W_1^*$. For every $i \in \left[n_M^k\right]$, if we replace all appearances of $\mathcal{W}_t^b$ in $\prod_{j=1}^{n_i^k} \mathrm{tr}\left(\widetilde{M}_{ij}^k\right) \widetilde{M}_{i0}^k$ and expand all parentheses of $\left(W_{t,2}^b W_{t,2}^b - W_2^* W_1^*\right)$, we have

$$\prod_{j=1}^{n_i^k} \mathrm{tr}\left(\widetilde{M}_{ij}^k\right) \widetilde{M}_{i0}^k = \sum_{l=1}^{2^d} \prod_{j=1}^{n_i^k} \mathrm{tr}\left(\widetilde{M}_{ij}^{kl}\right) \widetilde{M}_{i0}^{kl}, \tag{11}$$

where $\widetilde{M}_{ij}^{kl}$'s are multiplicative terms of parameter matrices $\left\{W_{t,1}^b, W_{t,2}^b\right\}$ and constant matrices $\{W_1^*, W_2^*\}$ such that

$$\sum_{j=0}^{n_i^k} \left(\deg\left(W_{t,1}^b; \widetilde{M}_{ij}^{kl}\right) + \deg\left(W_{t,2}^b; \widetilde{M}_{ij}^{kl}\right)\right) \leqslant 3d + \sum_{r=0}^{m} \left(\deg\left(W_{t,1}^b; M_r\right) + \deg(W_{t,2}^b; M_r)\right), \tag{12}$$

where the inequality comes from (9) and (10) and the fact that each $g_{t,1}^b$ or $g_{t,2}^b$ contributes 2 or 0 degrees in the form of $W_{t,2}^b W_{t,1}^b$ or $W_2^* W_1^*$, respectively.

Combining (7), (8) and (11), we have

$$\mathbb{E}\left[M\big|\mathcal{F}_t^b\right] = \sum_{k=1}^{n_M} \left(e_{k,d} + e_{k,d-1}\frac{1}{b} + \cdots + e_{k,0}\frac{1}{b^d}\right) \mathbb{E}\left[\widehat{M}_k\big|\mathcal{F}_t^b\right]$$

$$= \sum_{k=1}^{n_M} \left(e_{k,d} + e_{k,d-1}\frac{1}{b} + \cdots + e_{k,0}\frac{1}{b^d}\right) \sum_{i=1}^{n_M^{s_k}} \sum_{l=1}^{2^d} \prod_{j=1}^{n_i^k} \mathrm{tr}\left(\widetilde{M}_{ij}^{kl}\right) \widetilde{M}_{i0}^{kl}$$

$$= N_0 + N_1\frac{1}{b} + \cdots + N_d\frac{1}{b^d},$$

where

$$N_r = \sum_{k=1}^{n_M} e_{k,d-r} \left(\sum_{i=1}^{n_M^{s_k}} \sum_{l=1}^{2^d} \prod_{j=1}^{n_i^k} \mathrm{tr}\left(\widetilde{M}_{ij}^{kl}\right) \widetilde{M}_{i0}^{kl}\right). \tag{13}$$

Note that all constants in (13) are independent of $b$ and combining with (12), we have finished the proof.

$$\square$$

*Proof of Lemma 9.* Simply using the fact that $W_{t,i}^b = W_{t-1,i}^b - \alpha_t g_{t-1,i}^b, i = 1, 2$, if we replace each $W_{t,i}^b$ in the left-hand-side of (13) by $W_{t-1,i}^b - \alpha_t g_{t-1,i}^b$ and expand all the parentheses, then each $M_i, i \in [0:m]$ becomes the sum of $2^{d_i}$ multiplicative terms of parameter matrices $\left\{g_{t,1}^b, g_{t,2}^b\right\}$ and constant matrices $\left\{W_{t,1}^b, W_{t,2}^b, W_1^*, W_2^*\right\}$ with degree at most $d_i$. As a result, $\prod_{i=1}^{m} \mathrm{tr}\left(M_i\right) M_0$ becomes the sum of $2^d$ terms in the form of $\prod_{i=1}^{m} \mathrm{tr}\left(M_{ik}\right) M_{0k}$ where $\deg\left(M_{ik}\right) \leqslant 2^{d_i}$, and therefore $\sum_{i=0}^{m} \deg\left(M_{ik}\right) \leqslant \prod_{i=0}^{m} 2^{d_i} = d$. $\square$

*Proof of Theorem 3.* We use induction on $t$ to show this result. The base case of $t = 0$ it is the same as the statement in Lemma 8.

Suppose that the statement holds for $t \geqslant 0$, and we consider the case of $t + 1$. By Lemma 8, there exists a set of multiplicative terms $\left\{ M_{t+1,i,j}^k, i \in [m_{t+1,k}], j \in [0 : m_{t+1,k,i}], k \in [0 : d] \right\}$ of parameter matrices $\{W_{t+1,1}^b, W_{t+1,2}^b\}$ and constant matrices $\{W_1^*, W_2^*\}$ such that

$$\mathbb{E}\left[M \big| \mathcal{F}_{t+1}^b\right] = N_{t+1,0} + N_{t+1,1}\frac{1}{b} + \cdots + N_{t+1,d}\frac{1}{b^d}, \tag{14}$$

where $N_{t+1,k} = \sum_{i=1}^{m_{t+1,k}} \prod_{j=1}^{m_{t+1,k,i}} \mathrm{tr}\left(M_{t+1,i,j}^k\right) M_{t+1,i,0}^k, k \in [0 : d]$. Here $m_{t+1,k}, m_{t+1,k,i}$ are constants independent of $b$, and $\sum_{j=0}^{m_{t+1,k,i}} \deg\left(M_{t+1,i,j}^k\right) \leqslant 3d + d'$.

For each $i \in [m_{t+1,k}]$ and each $k \in [0 : d]$, by Lemma 9, there exists a set of multiplicative terms $\{M_{t,i,j,k,l}, j \in [m_{t+1,i,k}], l \in [d_{t,i,k}]\}$ of parameter matrices $\{g_{t,1}^b, g_{t,2}^b\}$ and constant matrices $\left\{W_{t,1}^b, W_{t,2}^b, W_1^*, W_2^*\right\}$ such that

$$\prod_{j=1}^{m_{t+1,k,i}} \mathrm{tr}\left(M_{t+1,i,j}^k\right) M_{t+1,i,0}^k = \sum_{l=1}^{d_{t,i,k}} \prod_{j=1}^{m_{t+1,k,i}} \mathrm{tr}\left(M_{t,i,j,k,l}\right) M_{t,i,0,k,l}, \tag{15}$$

where $d_{t,i,k} = 2^{\sum_{j=0}^{m_{t+1,k,i}} \left(\deg\left(W_{t,1}^b; M_{t,i,j,k,l}\right) + \deg\left(W_{t,2}^b; M_{t,i,j,k,l}\right)\right)}$ is a constant independent of $b$ and

$$\sum_{j=0}^{m_{t+1,k,i}} \deg\left(M_{t,i,j,k,l}\right) \leqslant 3d + d', \tag{16}$$

and

$$\sum_{j=0}^{m_{t+1,k,i}} \left(\deg\left(W_{t,1}; M_{t,i,j,k,l}\right) + \deg\left(W_{t,2}; M_{t,i,j,k,l}\right)\right) \leqslant 3d + d'. \tag{17}$$

Combining (14) and (15), we have for every $k \in [0 : d]$

$$N_{t+1,k} = \sum_{i=1}^{m_{t+1,k}} \sum_{l=1}^{d_{t,i,k}} \prod_{j=1}^{m_{t+1,k,i}} \mathrm{tr}\left(M_{t,i,j,k,l}\right) M_{t,i,0,k,l}. \tag{18}$$

Note that

$$\mathbb{E}\left[M|\mathcal{F}_0\right] = \mathbb{E}\left[\mathbb{E}\left[M\big|\mathcal{F}_{t+1}^b\right]\big|\mathcal{F}_0\right] = \mathbb{E}\left[N_{t+1,0}|\mathcal{F}_0\right] + \mathbb{E}\left[N_{t+1,1}|\mathcal{F}_0\right]\frac{1}{b} + \cdots + \mathbb{E}\left[N_{t+1,d}|\mathcal{F}_0\right]\frac{1}{b^d}$$

$$= \sum_{i=1}^{m_{t+1,0}} \sum_{l=1}^{d_{t,i,0}} \mathbb{E}\left[\prod_{j=1}^{m_{t+1,0,i}} \mathrm{tr}\left(M_{t,i,j,0,l}\right) M_{t,i,0,0,l}\bigg|\mathcal{F}_0\right] +$$

$$+ \sum_{i=1}^{m_{t+1,1}} \sum_{l=1}^{d_{t,i,1}} \mathbb{E}\left[\prod_{j=1}^{m_{t+1,1,i}} \mathrm{tr}\left(M_{t,i,j,1,l}\right) M_{t,i,0,1,l}\bigg|\mathcal{F}_0\right]\frac{1}{b} + \cdots +$$

$$+ \sum_{i=1}^{m_{t+1,d}} \sum_{l=1}^{d_{t,i,d}} \mathbb{E}\left[\prod_{j=1}^{m_{t+1,d,i}} \mathrm{tr}\left(M_{t,i,j,d,l}\right) M_{t,i,0,d,l}\bigg|\mathcal{F}_0\right]\frac{1}{b^d}, \tag{19}$$

and each $M_{t,i,j,k,l}$ is a multiplicative term of parameter matrices $\{g_{t,1}^b, g_{t,2}^b\}$ and constant matrices $\{W_{t,1}^b, W_{t,2}^b, W_1^*, W_2^*\}$ such that the degree is at most 1. Therefore, by induction, for every $i, k, l$, we have

$$\mathbb{E}\left[\prod_{j=1}^{m_{t+1,k,i}} \mathrm{tr}\left(M_{t,i,j,k,l}\right) M_{t,i,0,k,l}\bigg|\mathcal{F}_0\right] = N_{t,i,k,l,0} + N_{t,i,k,l,1}\frac{1}{b} + \cdots N_{t,i,k,l,q_t}\frac{1}{b^{q_t}}, \tag{20}$$

where $q_t \leqslant d' + \frac{1}{2}(3^t - 1)(3d + d')$ and $N_{t,i,k,l,0}, \cdots, N_{t,i,k,l,q_t}$ are sum of multiplicative terms of parameter matrices $\{W_{0,1}^b, W_{0,2}^b\}$ and constant matrices $\{W_1^*, W_2^*\}$ with degree at most $d \cdot 3^t$.

Combining (19) and (20), we can rewrite

$$\mathbb{E}\left[M|\mathcal{F}_0\right] = N_0 + N_1 \frac{1}{b} + \cdots + N_q \frac{1}{b^q},$$

in the same form as in the statement. Here $q \leqslant d + 3q_t \leqslant \frac{1}{2}(3^{t+2} - 1)d + \frac{1}{2}(3^{t+1} - 1)d'$ and $\sum_{j=0}^{m_{ki}} \deg\left(M_{ij}^k\right) \leqslant 3 \times 3^t(3d + d') = 3^{t+1}(3d + d')$ follow from (16) and (17).

In conclusion, we have shown that the statement holds for $t + 1$, and therefore finishes the proof.

$\square$

By changing the role of parameter and constant matrices in Theorem 3, we obtain the following corollary.

**Corollary 2.** *Given $t \geqslant 0$, for any multiplicative terms $M_i, i \in [0:m]$ of parameter matrices $\{W_{t,1}^b, W_{t,2}^b, \mathcal{W}_t^b\}$ and constant matrices $\{W_1^*, W_2^*\}$ such that $\sum_{i=1}^2 \deg\left(W_{t,i}^b; M\right) = d$ and $\deg\left(\mathcal{W}_t^b; M\right) = d'$, we denote $M = \prod_{i=1}^m \operatorname{tr}(M_i) M_0$. There exists a set of multiplicative terms $\left\{M_{ij}^k, i \in [m_k], j \in [0:m_{ki}], k \in [0:q]\right\}$ of parameter matrices $\{W_{0,1}^b, W_{0,2}^b\}$ and constant matrices $\{W_1^*, W_2^*\}$ such that*

$$\mathbb{E}\left[M|\mathcal{F}_0\right] = N_0 + N_1 \frac{1}{b} + \cdots + N_q \frac{1}{b^q},$$

*where $N_k = \sum_{i=1}^{m_k} \prod_{j=1}^{m_{ki}} \operatorname{tr}\left(M_{ij}^k\right) M_{i0}^k, k \in [0:q]$. Here $m_k, m_{ki}$ and $q \leqslant 3^t(d + 2d')$ are constants independent of $b$, and $\sum_{j=0}^{m_{ki}} \deg\left(M_{ij}^k\right) \leqslant 3^t(d + 2d')$.*

*Proof of Corollary 2.* We simply note that $M$ can be written as the sum of at most $2^d$ multiplicative terms of parameter matrices $\{W_{t,1}^b, W_{t,2}^b, W_1^*, W_2^*\}$ and constant matrix $\{I_0\}$. Then we apply Lemmas 8 and 9 iteratively in the same way as in the proof of Theorem 3 to finish the proof. $\square$

*Proof of Theorem 4.* We only show the case for $g_{t,1}$ since the proof for $g_{t,2}$ can be tackled similarly. Note that

$$\operatorname{var}\left(g_{t,1}^b|\mathcal{F}_0\right) = \operatorname{var}\left(\frac{1}{b}\sum_{i=1}^b W_{t,2}^b{}^T \mathcal{W}_t^b x_{t,i} x_{t,i}^T \Big| \mathcal{F}_0\right) = \frac{1}{b^2}\sum_{i=1}^b \operatorname{var}\left(W_{t,2}^b{}^T \mathcal{W}_t^b x_{t,i} x_{t,i}^T \Big| \mathcal{F}_0\right)$$

$$= \frac{1}{b}\operatorname{var}\left(W_{t,2}^b{}^T \mathcal{W}_t^b x_{t,1} x_{t,1}^T \Big| \mathcal{F}_0\right)$$

$$= \frac{1}{b}\left(\mathbb{E}\left[\left\|W_{t,2}^b{}^T \mathcal{W}_t^b x_{t,1} x_{t,1}^T\right\|^2 \Big| \mathcal{F}_0\right] - \left\|\mathbb{E}\left[W_{t,2}^b{}^T \mathcal{W}_t^b x_{t,1} x_{t,1}^T \Big| \mathcal{F}_0\right]\right\|^2\right)$$

$$= \frac{1}{b}\left(\mathbb{E}\left[\operatorname{tr}\left(x_{t,1} x_{t,1}^T \mathcal{W}_t^b{}^T W_{t,2}^b W_{t,2}^b{}^T \mathcal{W}_t^b x_{t,1} x_{t,1}^T\right) \Big| \mathcal{F}_0\right] - \left\|\mathbb{E}\left[W_{t,2}^b{}^T \mathcal{W}_t^b x_{t,1} x_{t,1}^T \Big| \mathcal{F}_0\right]\right\|^2\right)$$

$$= \frac{1}{b}\left(\mathbb{E}\left[\mathbb{E}\left[\operatorname{tr}\left(x_{t,1} x_{t,1}^T \mathcal{W}_t^b{}^T W_{t,2}^b W_{t,2}^b{}^T \mathcal{W}_t^b x_{t,1} x_{t,1}^T\right) \Big| \mathcal{F}_t^b\right] \Big| \mathcal{F}_0\right] - \left\|\mathbb{E}\left[\mathbb{E}\left[W_{t,2}^b{}^T \mathcal{W}_t^b x_{t,1} x_{t,1}^T \Big| \mathcal{F}_t^b\right] \Big| \mathcal{F}_0\right]\right\|^2\right)$$

$$= \frac{1}{b}\left(\mathbb{E}\left[(p + 2)\operatorname{tr}\left(\mathcal{W}_t^b{}^T W_{t,2}^b W_{t,2}^b{}^T \mathcal{W}_t^b\right) \Big| \mathcal{F}_0\right] - \left\|\mathbb{E}\left[W_{t,2}^b{}^T \mathcal{W}_t^b \Big| \mathcal{F}_0\right]\right\|^2\right)$$

$$= \frac{1}{b}\left((p + 2)\operatorname{tr}\left(\mathbb{E}\left[\mathcal{W}_t^b{}^T W_{t,2}^b W_{t,2}^b{}^T \mathcal{W}_t^b \Big| \mathcal{F}_0\right]\right) - \left\|\mathbb{E}\left[W_{t,2}^b{}^T \mathcal{W}_t^b \Big| \mathcal{F}_0\right]\right\|^2\right).$$

$$= \frac{1}{b}\left((p + 2)\operatorname{tr}\left(\mathbb{E}\left[\mathcal{W}_t^b{}^T W_{t,2}^b W_{t,2}^b{}^T \mathcal{W}_t^b \Big| \mathcal{F}_0\right]\right) - \left\|\mathbb{E}\left[W_{t,2}^b{}^T \mathcal{W}_t^b \Big| \mathcal{F}_0\right]\right\|^2\right).$$

Here we have used the fact that $\mathbb{E}_{x \sim \mathcal{N}(0, I_p)} \operatorname{tr}\left(xx^T A xx^T\right) = (p+2)\operatorname{tr}(A)$. By Corollary 2 we know that there exists a set of multiplicative terms $\left\{M_{ij}^k, i \in [m_k], j \in [0:m_{ki}], k \in [0:q]\right\}$ of parameter matrices $\{W_{0,1}^b, W_{0,2}^b\}$ and constant matrices $\{W_1^*, W_2^*\}$ such that

$$\operatorname{tr}\left(\mathbb{E}\left[\mathcal{W}_t^b{}^T W_{t,2}^b W_{t,2}^b{}^T \mathcal{W}_t^b \Big| \mathcal{F}_0\right]\right) = \gamma_0 + \gamma_1 \frac{1}{b} + \cdots + \gamma_q \frac{1}{b^q}, \tag{21}$$

where $\gamma_k = \sum_{i=1}^{m_k} \prod_{j=0}^{m_{ki}} \mathrm{tr}\left(M_{ij}^k\right), k \in [0:q]$. Here $m_k, m_{ki}$ and $q \leqslant 6 \cdot 3^t$ are constants independent of $b$, and $\sum_{j=0}^{m_{ki}} \deg\left(M_{ij}^k\right) \leqslant 6 \cdot 3^t$. Note that $W_{0,1}^b, W_{0,2}^b$ are fixed, and we have $\gamma_k, k \in [0:q]$ are constants independent of $b$.

Similarly we observe that there exist constants $q' \leqslant 2 \cdot 3^{t+1}$ and $\gamma'_k, k \in [0:q']$ such that

$$\left\| \mathbb{E}\left[ W_{t,2}^{b}{}^T \mathcal{W}_t^b \middle| \mathcal{F}_0 \right] \right\|^2 = \gamma'_0 + \gamma'_1 \frac{1}{b} + \cdots + \gamma'_q \frac{1}{b^{q'}}. \tag{22}$$

By defining $\gamma_i = 0, i > q$ and $\gamma'_i = 0, i > q'$, and combining (21) and (22) we have

$$\mathrm{var}\left(g_{t,1}^b \middle| \mathcal{F}_0\right) = \frac{1}{b}\left( (p+2)\mathrm{tr}\left(\mathbb{E}\left[\mathcal{W}_t^{b\,T} W_{t,2}^b W_{t,2}^{b}{}^T \mathcal{W}_t^b \middle| \mathcal{F}_0\right]\right) - \left\|\mathbb{E}\left[W_{t,2}^{b}{}^T \mathcal{W}_t^b \middle| \mathcal{F}_0\right]\right\|^2 \right)$$

$$= \frac{p+2}{b}\left(\gamma_0 + \gamma_1 \frac{1}{b} + \cdots + \gamma_q \frac{1}{b^q}\right) - \frac{1}{b}\left(\gamma'_0 + \gamma'_1 \frac{1}{b} + \cdots + \gamma'_q \frac{1}{b^{q'}}\right)$$

$$= \sum_{k=1}^{\max\{q,q'\}} \left((p+1)\gamma_k - \gamma'_k\right)\frac{1}{b^k}.$$

Note that $\gamma_k$'s and $\gamma'_k$'s are all constants independent of $b$, and $\max\{q, q'\} \leqslant 2 \cdot 3^{t+1}$. This completes the proof.

$\square$

*Proof of Theorem 5.* We first show that in

$$\mathrm{var}\left(g_{t,i}^b \middle| \mathcal{F}_0\right) = \beta_1 \frac{1}{b} + \cdots + \beta_r \frac{1}{b^r}$$

we have $\beta_1 \geqslant 0$. If $r = 1$, the statement obviously holds. Let us assume that the statement does not hold for $r > 1$, i.e. $\beta_1 < 0$. Taking $b$ large enough such that $\beta_1 b^{r-1} + \beta_2 b^{r-2} + \cdots + \beta_r < 0$ yields

$$\mathrm{var}\left(g_{t,i}^b \middle| \mathcal{F}_0\right) = \frac{1}{b^r}\left(\beta_1 b^{r-1} + \beta_2 b^{r-2} + \cdots + \beta_r\right) < 0,$$

which contradicts the fact that $\mathrm{var}\left(g_{t,i}^b \middle| \mathcal{F}_0\right) \geqslant 0$. Therefore, we have $\beta_1 \geqslant 0$.

Let $b_0$ be large enough such that for all $b \geqslant b_0$, we have $\beta_1 b^{r-1} + 2\beta_2 b^{r-2} + \cdots + r\beta_r \geqslant 0$. We denote $f(b) = \beta_1 \frac{1}{b} + \beta_2 \frac{1}{b^2} + \cdots + \beta_r \frac{1}{b^r} \geqslant 0$. For all $b > b_0$ we have

$$f'(b) = -\frac{1}{b^{r+1}}\left(\beta_1 b^{r-1} + 2\beta_2 b^{r-2} + \cdots + r\beta_r\right) \leqslant 0.$$

Therefore, for all $b > b_0$ we have $\left(\mathrm{var}\left(g_{t,i}^b \middle| \mathcal{F}_0\right)\right)' = -\frac{r}{b^{r+1}}f(b) + \frac{1}{b^r}f(b) \leqslant 0$, and thus $\mathrm{var}\left(g_{t,i}^b \middle| \mathcal{F}_0\right)$ is a decreasing function of $b$ for all $b > b_0$.

$\square$

## B.3 EXTENSION TO DEEP LINEAR NETWORKS

The extension from two-layer linear network to deep linear network is straightforward. Here we only provide the ideas on how to translate the proof of two-layer network to $d$-layer network, but not the strict proof. For simplicity, we remove all superscripts $b$ of matrices in this subsection.

Assume that the $d$-layer linear network is given by $f(x; w) = W_d W_{d-1} \cdots W_2 W_1 x$, where $W_i, i \in [d]$ is the parameter matrix on the $i$-th layer and $w = (W_1, \ldots, W_d)$. The population loss is defined as

$$\mathcal{L}(w) = \mathbb{E}_{x \sim \mathcal{N}(0, I_p)}\left[\frac{1}{2}\|W_d \cdots W_1 x - W_d^* \cdots W_1^* x\|^2\right].$$

Similar to (1) and (2), we have

$$g_{t,k} = \frac{1}{b}\sum_{i=1}^b \nabla_{W_{t,k}}\left(\frac{1}{2}\|W_{t,d} \cdots W_{t,1} x_{t,i} - W_d^* \cdots W_1^* x_{t,i}\|^2\right)$$

$$= \frac{1}{b}\sum_{i=1}^b W_{t,k+1}^T \cdots W_{t,d}^T \left(W_d \cdots W_1 - W_d^* \cdots W_1^*\right) x_{t,i} x_{t,i}^T W_{t,1}^T \cdots W_{t,k-1}^T, \quad k \in [d].$$

We denote $\mathcal{W}_t = W_{t,d} \cdots W_{t,1} - W_d^* \cdots W_1^*$. The remaining are all the same as the proofs in Appendix B.2, except we should replace all appearance of $\{W_{t,2}, W_{t,1}\}$ to $\{W_{t,d}, W_{t,d-1}, \cdots, W_{t,1}\}$ and all $\{W_2^*, W_1^*\}$ to $\{W_d^*, W_{d-1}^*, \cdots, W_1^*\}$. We can do this because the stochastic gradient $g_{t,k}$ is still the sum of multiplicative terms of parameter matrices $\{x_{t,i}\}$ and constant matrices $\{W_{t,d}, \cdots, W_{t,1}, W_d^*, \cdots, W_1^*\}$ so the Lemmas in Appendix B.2 still apply.

In conclusion, we can again represent $\mathrm{var}\,(g_{t,k}|\mathcal{F}_0), k \in [d]$ as a polynomial of $\frac{1}{b}$ with finite degree and without the constant term. By the same approach in the proof of Theorem 5, we can show that the variance is a decreasing function of the mini-batch size $b$.

