# OpenReview forum: "The Impact of the Mini-batch Size on the Dynamics of SGD: Variance and Beyond"
_ICLR.cc/2021/Conference — Reject_

### Official Review · AnonReviewer3 · 2020-10-25
**Minor refinement of the estimation of stochastic gradient**

**Rating:** 3
**Confidence:** 5

**Review:**

This paper studies the variance of stochastic gradient in SGD conditioned on the initialization point. It shows that the variance of stochastic gradient is a decreasing function of minibatch size for linear regression and deep linear network. Compared with previous works that show similar the results for one step in SGD, the results in this work only rely on initialization point.
Although the technique in this paper is different from previous works,  the setting in this paper can only handle linear model which is simple and limited. Besides, only showing that the tendency of gradient variance as minibatch size changes is not enough,  and it is better to give the decreasing rate. So the impact of the refinement on the estimation of gradient variance is not clearly stated. If the authors can provide some examples to show that the analyses in previous works are too coarse to give the wrong direction, the impact of this paper can be enhanced. In current version of the paper, I can not get any new insights from the theoretical and experimental results because there have been many existing works that can tell the relation (or tendency) between gradient variance and minibatch.

Detailed comments:
1. The notation "var()" is not rigorous because the gradient is high-dimensional. It should be replaced by the covariance matrix of a random vector.  Given that the gradient is a high-dimensional vector and the "variance" is a matrix, it is not clear that what the meaning of "the variance is a decreasing function of minibatch size" is. Does it mean that all the diagonal elements in the covariance matrix is a decreasing function?
2. It must be clarified that which distribution the expectation and variance are taken over. For example, var(\nabla L(w_t)) should be written as var_{w_t}(\nabla L(w_t)).
3. In the setting of deep linear network, the distribution of data is assumed to be normal distribution, which is strong. Is this condition necessary in the proof?
4. The experiments are not sufficient to support the claim. Can you provide results for more selections of minibatch sizes, more datasets and more NN architectures?
5. In future works, the authors point out many interesting research problems, but it is not clear that why the techniques and results in this work can help to solve those problems. Can you provide connections between the main results and the future work? For example, which technique can help design better variance reduction method? Clarifying this will be helpful to evaluate the impact of this paper.

---

> ### Author Response · Authors · 2020-11-21
> **Response to R3**
>
> We appreciate Reviewer3’s detailed feedback. R1's review does not give credit to several key contributions in the paper-which we think there could be a misunderstanding. We address R1's review in details as below.
>
> > the setting in this paper can only handle linear model which is simple and limited.
>
> We believe the linear models are not simple and limited; but in fact, are vast in the literature and can provide precious insights for further research. More details are addressed in the common rebuttal.
>
> > Besides, only showing that the tendency of gradient variance as minibatch size changes is not enough, and it is better to give the decreasing rate. So the impact of the refinement on the estimation of gradient variance is not clearly stated.
>
> The decreasing rate is exactly $\mathcal{O}(\frac{1}{b})$, as we clearly present in Theorem 4. We are not sure if the reviewer is referring to the rate in terms of $t$. If this is the case, we can achieve this by referrign to the properties of each coefficient $\beta_i$ in the polynomial representation of the variance. Estimations of the orders of these coefficients are given at the top of page 6.
>
> > If the authors can provide some examples to show that the analyses in previous works are too coarse to give the wrong direction, the impact of this paper can be enhanced. In current version of the paper, I can not get any new insights from the theoretical and experimental results because there have been many existing works that can tell the relation (or tendency) between gradient variance and minibatch.
>
> Existing literature mostly focus on the ''one-step'' variance of the stochastic gradient estimator, while our paper focuses on the ''$t$-step'' variance. While the prior ''one-step'' results are relevant, our ''$t$-step'' results are much more aligned with practice and the proof techniques are vastly different from those used in ''one-step''.
> In fact, our results align with the common intuition that the variance of the SG estimators is decreasing with respect to the mini-batch size. We refine this common sense to ``$t$-step'' case and prove it. To the best of the authors' knowledge, this paper is the first result to mathematically prove this statement.
>
> > The notation "var()" is not rigorous because the gradient is high-dimensional. It should be replaced by the covariance matrix of a random vector. Given that the gradient is a high-dimensional vector and the "variance" is a matrix, it is not clear that what the meaning of "the variance is a decreasing function of minibatch size" is. Does it mean that all the diagonal elements in the covariance matrix is a decreasing function?
>
> We define the ''scalar'' variance at the beginning of Section 3. This is a common practice in analyzing the dynamics of the SGD algorithm (e.g. equation (4.6) in [Bottou et al. 2018]). We will update the next version of the paper to further clarify this.
>
> > It must be clarified that which distribution the expectation and variance are taken over. For example, $var(\nabla L(w_t))$ should be written as $var_{w_t}(\nabla L(w_t))$.
>
> Throughout the paper, the expectations and variances are taken over all possible choices of the mini-batches at each time step. We mention this in the second paragraph of Section 3 but we agree with the reviewer that we should clarify it more clearly. This will be updated in the next version of the paper. Writing $\textbf{Var}_{w_t}(\nabla L(w_t))$ could be misleading about where the randomness comes from.
>
> > In the setting of deep linear network, the distribution of data is assumed to be normal distribution, which is strong. Is this condition necessary in the proof?
>
> Yes this is necessary in the proof. The whole proof of Section 3.2 is based on the normal assumption so that we can use Lemma 7 in the appendix to build up the bridge between $g_t^b$ and $w_t^b$. We agree with the reviewer that this might be a strong assumption for deep learning practice. However, it is common in many theoretical results, e.g. in [Li et al. 2017, Ge et al. 2017, Soltanolkotabi et al. 2018]. For more details about how to get rid of the normal assumption, please refer to the common response.
>
> > The experiments are not sufficient to support the claim. Can you provide results for more selections of minibatch sizes, more datasets and more NN architectures?
>
> Since we focus on variance, a single experiment requires many runs leading to significant time to conduct them. We have done our best to keep them under control and yet to convey the most important ideas.
>
> > In future works, the authors point out many interesting research problems, but it is not clear that why the techniques and results in this work can help to solve those problems. Can you provide connections between the main results and the future work? For example, which technique can help design better variance reduction method? Clarifying this will be helpful to evaluate the impact of this paper.
>
> Addressed in the common part.

---

> > ### Author Response · Authors · 2020-11-21
> > **References**
> >
> > Reference
> >
> > - Bottou, Léon, Frank E. Curtis, and Jorge Nocedal. "Optimization methods for large-scale machine learning." Siam Review 60.2 (2018): 223-311.
> >
> > - Li, Yuanzhi, and Yang Yuan. "Convergence analysis of two-layer neural networks with relu activation." Advances in neural information processing systems. 2017.
> >
> > - Ge, Rong, Jason D. Lee, and Tengyu Ma. "Learning one-hidden-layer neural networks with landscape design." arXiv preprint arXiv:1711.00501 (2017).
> >
> > - Soltanolkotabi, Mahdi, Adel Javanmard, and Jason D. Lee. "Theoretical insights into the optimization landscape of over-parameterized shallow neural networks." IEEE Transactions on Information Theory 65.2 (2018): 742-769.

---

### Official Review · AnonReviewer2 · 2020-10-26
**reject**

**Rating:** 4
**Confidence:** 4

**Review:**

This paper studied mini-batch stochastic gradient descent (SGD) for linear regression and linear neural networks. The analysis show that the variance of the stochastic gradient estimator is a decreasing function of b (mini-batch size). The authors also empirically validate their observations on MNIST and YELP review datasets.

I think the main result of this paper is not interesting. There are some comments.

1. The connection between mini-batch size and the variance of gradient estimator almost will be considered in any paper about SGD and I do not think this paper provide any novel perspective.
2. The analysis in section 3.2 discuss the extension on linear network, however, we are more interested in nonlinear case. The current analysis looks cannot be extended to nonlinear model directly.
3. The experimental section also should be improved. The structure of the network should be introduced in details. It is prefer to validate the proposed theory on more difficult task. MNIST and YELP cannot reflect the power of deep learning model.

---

> ### Author Response · Authors · 2020-11-21
> **Response to R2**
>
> We thank Reviewer2 for the valuable feedback. We here address the comments and will incorporate all the feedback.
>
> > The connection between mini-batch size and the variance of gradient estimator almost will be considered in any paper about SGD and I do not think this paper provide any novel perspective.
>
> We emphasize that all the results in our paper are regarding only fixing the initial weights, but not restricted to a given iteration (which existing literature focuses on).
>
> To the best of the authors' knowledge, this result has not yet been proven, or it can not be naturally deduced by any existing textbook or literature results. In fact, it is non-trivial and it requires a sophisticated mathematical proof. These proof techniques are exactly the main contribution of this paper. If the reviewer is aware of any related literature or believes this result is not novel, we suggest s/he provides detail information about the relevant literature or proof. We would very welcome such pointers.
>
>
> > The analysis in section 3.2 discuss the extension on linear network, however, we are more interested in nonlinear case. The current analysis looks cannot be extended to nonlinear model directly.
>
> We agree with the reviewer that the results in non-linear case are more interesting, but the current results are also valuable. We answered this in details in the general rebuttal.
>
> > The experimental section also should be improved. The structure of the network should be introduced in details. It is prefer to validate the proposed theory on more difficult task. MNIST and YELP cannot reflect the power of deep learning model.
>
> Given the space limit, we could not put the introduction of the datasets and models in the main body of the paper. They are given in detail in Appedix A. We agree with the reviewer that MNIST and YELP may not be the best candidates for evaluating deep learning models. The issue is that we need to perform many runs of SGD from the same initial points in order to achieve appropriate confidence intervals which is computationally very expensive. For this reason, we decided not use large datasets and models that would require months of GPU time.

---

### Official Review · AnonReviewer4 · 2020-10-27
**Review of the impact of mini-batch sgd size on...**

**Rating:** 6
**Confidence:** 4

**Review:**

## Summary and contributions

This paper tackles the problem of the impact of mini-batch size on the variance of the gradients of SGD. Unlike most work on the topic, it does not study the variance of the gradient conditionned over the last iteration, but rather the absolute variance of the gradient conditioned only on the initial point.

The paper restricts itself to linear models, either with least mean square regression or deep linear model with 2 layers (which would be non convex but still linear).
The paper shows two results. For linear models, the variance of the gradients conditionned on the initial point is decreasing with the batch size b.
For a deep linear model, the result shows that the variance of gradient is a polynomial in 1/b with no constant term. Therefore, for b >= b0 for some b0, the variance is decreasing with b.

## Review

This paper tackles the problem of the variance in SGD from a novel angle, namely the total variance conditioning only on the initial weights, and not on the previous iteration as usually done.

The result on linear regression seems very natural, and the proof is done elegantly (I checked up to the proof of Theorem 1).

I haven't checked the proof for the deep linear network. The result is more interesting than in the linear regression case because the 2 layers linear network is non convex, and therefore, one can imagine having multiple local minima and 2 different trajectory starting from the same point to diverge at some point, which could lead to drastically different gradients. This is not really possible for least mean square with a full rank hessian, as there is a single optimum, and in any case all trajectories will end up at the same place.

Having a small total variance conditioned only on the initial point means that somehow the trajectories for different samplings cannot diverge too much. Of course, if the batch size goes to infinity, one is doing gradient descent, and all the trajectories are exactly the same which is compatible with theorem 4. I think the proof technique is interesting to be able to bound or study deviations between trajectories. However, in its current form it is non practical as the bound is very complex and in particular can increase as the iteration increases. In essence, the results says that for any iteration, as B increases, trajectories get closers, but it does not say that they will stay close for any iteration number.

I must say that while the theoretical part is doing a good job, the experiment part in Section 4 is quite poor.
The author tries to open up to the problem of generalization.
Speaking of figure 1 b), the authors say that the validation loss improves with the batch size, i.e. the generalization ability is better with large batch sizes. This however contradicts previous work that have been mentioned by the authors in their very introduction. The authors do not comment on this contradiction. My own conclusion is that deriving general results on generalization from MNIST is a perilous exercice, if not plain wrong.


## Conclusion

Overall, I would say that this paper is just above the acceptance bar, because the theory holds up well and could be of interest for finer analysis of the dynamics of SGD, and in particular of different trajectory starting from the same point (how quickly will they diverge?), although doing so would require significant extension to the present work.
The authors try to open up to the problem of generalisation but fail to a proper theoretical link with their own work, while their experimental results are obtained only on MNIST and therefor subject to caution (and in fact contradict previous work).


## Notes and remarks

- In the abstract, the authors write "for deep neural network with L2 loss, we show...". This is not true, the result is only for two layers deep linear network.
- Definition 1: the definition of the degree is a bit hand wavy, there should be some minimum over all possible decomposition of M. This is especially needed if for some X, both X and X^{-1} belong to \mathcal{X}.
- page 18, at the top: "We denote A = \sum ...", A has already been introduced before end of page 17, but with x_i x_i^T instead of C_i (both are equal of course).
- Section 4, comments on the loss of Figure 1: phrasing "loss is decreasing with b" is ambigous or wrong. For the training, the loss increases with the batch size. I guess the authors meant that the loss worsen with b (as low is good), but worsen != decreasing.

---

> ### Author Response · Authors · 2020-11-21
> **Response to R4**
>
> We thank Reviewer4 for the valuable feedback and for appreciating our contribution. We address below the raised concerns.
>
>
> > I think the proof technique is interesting to be able to bound or study deviations between trajectories. However, in its current form it is not practical as the bound is very complex and in particular can increase as the iteration increases. In essence, the results says that for any iteration, as B increases, trajectories get closers, but it does not say that they will stay close for any iteration number.
>
> We thank the reviewer for carefully checking the technical details of the paper. The dynamics of neural network training are extremely complicated, and only given the initial point, calculating any quantity at time $t$ definitely takes exponential efforts. We assume the reviewer's ''stay close for any iteration'' is aiming to ask at what iteration, the variance will be smaller than a given threshold. This can be partially answered from the existing results in the paper where we give orders of the coefficients in the polynomial representation of the variance. However, a more detailed and precise analysis on the coefficients is needed. This could be a future work of this paper.
>
> > I must say that while the theoretical part is doing a good job, the experiment part in Section 4 is quite poor. The author tries to open up to the problem of generalization. Speaking of figure 1 b), the authors say that the validation loss improves with the batch size, i.e. the generalization ability is better with large batch sizes. This however contradicts previous work that have been mentioned by the authors in their very introduction. The authors do not comment on this contradiction. My own conclusion is that deriving general results on generalization from MNIST is a perilous exercice, if not plain wrong.
>
> We believe that smaller mini-batch sizes lead to better generalization ability and the experiments also verify this. We are not using validation loss (nor the gap between training and validation losses) as the indicator of generalization. In fact, we build a distorted test set of MNIST following [Simard et al. 2003] and measure model's generalization ability by the gap of accuracy on test/training sets. More details are available in the second paragraph in page 16 and Figure 1(d) or 4(b).
>
> We agree with the reviewer that MNIST is probably not the best dataset to study the accuracy/generalization ability. The issue is that we need to perform many runs in order to achieve appropriate confidence intervals which is computationally very expensive. For this reason, we decided to use MNIST and not larger datasets that would require months of GPU time. We focus the content on theoretical aspects and the generalization versus mini-batch size is given less importance.
>
> > Minor problems:
> > - In the abstract, the authors write "for deep neural network with L2 loss, we show...". This is not true, the result is only for two layers deep linear network.
> > - page 18, at the top: "We denote $A = \sum$ ...", A has already been introduced before end of page 17, but with $x_i x_i^T$ instead of $C_i$ (both are equal of course).
> > - Section 4, comments on the loss of Figure 1: phrasing "loss is decreasing with b" is ambigous or wrong. For the training, the loss increases with the batch size. I guess the authors meant that the loss worsen with b (as low is good), but worsen != decreasing.
>
> We have addressed these remarks in the revision. We thank the reviewer for carefully proofreading and pointing out these aspects.
>
> > Definition 1: the definition of the degree is a bit hand wavy, there should be some minimum over all possible decomposition of M. This is especially needed if for some X, both $X$ and $X^{-1}$ belong to $\mathcal{X}$.
>
> We agree with the reviewer that the definition is slightly loose. We will address this in the revision.
>
> ### References
>
> - Simard, Patrice Y., David Steinkraus, and John C. Platt. "Best practices for convolutional neural networks applied to visual document analysis." Icdar. Vol. 3. No. 2003. 2003.

---

### Official Review · AnonReviewer1 · 2020-10-28
**Official Blind Review #1**

**Rating:** 5
**Confidence:** 4

**Review:**

The paper shows that the variance of the gradient has an inverse dependence on the batch size in linear networks, subject to the knowledge of the initial weights. The main novelty of the paper is the computation of an exact dependence between batch size and variance of the gradient in the linear regression setting. In addition to that, the authors conduct a lot of experiments involving non-linear networks and real-world datasets, that show the inverse dependence of the variance of gradient and batch size throughout the training.


My major concern is that the authors don't provide an application of their theorems, i.e. a setting where the exact knowledge of the variance of the gradient is useful. It will be useful to know if their theorems help tighten the convergence rates of mini-batch SGD in convex regression or improve generalization bounds for mini-batch SGD. Or can the authors say that their theorems help to tune the learning rate when going from large batch SGD to small-batch SGD? Without any such application, I believe the theorems are incomplete.

My other concerns are the following:
1. The authors prove that the variance is inversely dependent on the batch size. However, their theorems also imply the variance of the gradient keeps on increasing with time. Hence, a convergence theorem with an appropriate learning rate decay will be helpful. I see in the experiments the authors have used the learning rate as $1/t$. A justification will be really helpful.
2. Most of the experiments in the paper have been conducted on non-linear networks and non-gaussian data (like MNIST and YELP). Hence, a rough idea about how the theorems can be extended to ReLU networks, even in the over parametrized case, will be highly appreciated.
3. The experiments show that small-batch SGD can reach lower training loss. Can the authors show why the training loss decreases with increasing variance in gradient, at least in the linear regression case?

My scores are on the lower side mainly because I believe the authors need to show some application of their theorems. I am happy to discuss this with the authors and other reviewers during the discussion period.

Minor concern:
The abstract and introduction have writing issues. However, I haven't taken them into account in my score. I would request the authors to improve those sections soon.

---

> ### Author Response · Authors · 2020-11-21
> **Response to R1**
>
> We thank Reviewer1 for the valuable feedback and for appreciating our contribution. We address below the raised concerns.
>
> > My major concern is that the authors don't provide an application of their theorems, i.e. a setting where the exact knowledge of the variance of the gradient is useful. It will be useful to know if their theorems help tighten the convergence rates of mini-batch SGD in convex regression or improve generalization bounds for mini-batch SGD. Or can the authors say that their theorems help to tune the learning rate when going from large batch SGD to small-batch SGD? Without any such application, I believe the theorems are incomplete.
>
> Addressed in general rebuttal.
>
> > The authors prove that the variance is inversely dependent on the batch size. However, their theorems also imply the variance of the gradient keeps on increasing with time. Hence, a convergence theorem with an appropriate learning rate decay will be helpful. I see in the experiments the authors have used the learning rate as $1/t$. A justification will be really helpful.
>
> We do not present a result (nor try to convey the idea) regarding the increasing variance of gradient over time. In fact, as we choose learning rate as $1/t$, the variance of the SG estimators finally converge from the experiments. We wonder what part of the theorems is misleading and leads the reviewer to conclude that the variance of the SG estimator is increasing as $b$ grows. We would like to improve the exposition of the paper.
>
> > Most of the experiments in the paper have been conducted on non-linear networks and non-gaussian data (like MNIST and YELP). Hence, a rough idea about how the theorems can be extended to ReLU networks, even in the over parametrized case, will be highly appreciated.
>
> Addressed in the general rebuttal.
>
> > The experiments show that small-batch SGD can reach lower training loss. Can the authors show why the training loss decreases with increasing variance in gradient, at least in the linear regression case?
>
> In linear regression case, we are able to show the closed formula of loss at time $t$ with respect to the mini-batch size $b$ iteratively. The decreasing property is a special case of that relationship. In the deep linear networks, the above statement still holds, although we could only give a polynomial representation instead of the closed form.
>
> This can be understood intuitively in the following way. Given $L_2$ loss, the variance of the (stochastic) gradient is a function of the parameter $w_t$ with order 2, and the loss also has order 2 (e.g. in linear regression, $\nabla L(w) = \frac{1}{n}\sum_{i=1}^n x_i (w^T x_i - y_i) $ and $L(w) = \frac{1}{2n} \sum_{i=1}^n \left(w^T x_i - y_i\right)^2$). As our techniques can take care of change in $w$ to any degree (e.g. Theorem 3), the discussion of loss is similar to the variance. In addition, our techniques can even calculate/estimate the variance of the loss, which is twice in the degree of $w$.

---

> > ### Comment · AnonReviewer1 · 2020-11-23
> > **Thanks for your response.**
> >
> > Thanks for your response. I strongly agree with the authors that deep linear networks are interesting models to work on and the results can provide insight into the training landscape of deep non-linear networks.
> >
> > However, I still feel a theorem stating an application of the current results will help strengthen the paper. Hence, I believe the authors should add one theorem stating such an application, probably one on the new convergence rates of linear classifiers with the new estimations for $M$ and $M_V$.

---

### Author Response · Authors · 2020-11-21
**General Rebuttal**

We thank the reviewers for their thoughtful comments. We answer three common questions here and leave the detailed feedback to separate responses.

## Contribution of the paper and application of the theorems

The main contributions of this paper are the tools provided to calculate and represent the variance of SG estimators in the $t$-th step (and not a single step), which is non-trivial and requires a sophisticated mathematical proof. These tools can be extended to calculate any form that has a polynomial relationship to the model parameters $w_t$, e.g. expectation/variance of the loss function, norm of the SG estimator to any degree.

We discuss the possible applications of our theorems in Section 5. One possible direction is to help tighten the convergence rates of SGD. Current analyses of SGD convergence rely on two constants $M$ and $M_V$ such that \begin{align*}
    \textrm{Var}\left(g_t^b\right) \le M + M_V \left\| \nabla L(w_t^b) \right\|^2.
\end{align*}
But it is unclear what are the exact values of $M$ and $M_V$ (see Assumption 4.3 of [Bottou et al. 2018] and the context therein). Even the exact values of $w_t$ nor $\nabla L(w_t)$ are not easily accessible at the beginning of training. It is a common practice to take relatively large $M$ and $M_V$ to make sure the above bound hold. However, this leads to a relatively poor convergence rate of the SGD algorithm. Our theorems are able to estimate these quantities and therefore provide finer analyses of the convergence of SGD.

On the other hand, many existing works focus on the relationship between the variance and the generalization bounds of SGD. Suppose that we have a relationship between the generalization ability and the variance $g = f(v)$, where $g$ is generalization and $v$ is the variance (this function should also depend on many other variables, but for simplicity we ignore them here). Given our relationship of the variance and the mini-batch size $b$, we can come up with an estimation $g = h(b)$. The function $f$ is beyond the scope of this paper but the authors are interested in developing such relationships as a follow-up work.

## Results on linear networks are simple and limited

Linear networks are common in theoretical deep learning studies. These networks are non-convex and require sophisticated mathematical tools to analyze their properties, and therefore the results around them provide precious insights on general non-convex deep learning models. Many works focus on the linear network models and have already shown huge impact in the community [Kawaguchi 2016, Arora et al. 2018, Laurent et al. 2017]. Another line of research [Jacob et al. 2018, Du et al. 2018, Arora et al. 2019, Lee et al. 2019] which focus on the convergence/generalization of non-linear neural networks usually assume the networks are heavily over-parameterized (or even have infinite width), which simplifies the networks to linear models by the neural tangent kernel approach. In conclusion, the study on deep linear networks is vast and is precious for further research.

---

> ### Author Response · Authors · 2020-11-21
> **General Rebuttal (continued)**
>
> ## Extension to non-linear cases and non-Gaussian data
>
> We highly agree with the reviewer that the theorems are much more exciting on non-linear networks or general data. This is also the interest of the authors as a subsequent work.
>
> For the general neural networks without any assumption on model capacity, we can approximate the non-linear activation functions by polynomials up to any precision. For example, suppose we approximate the Sigmoid function $\sigma(x)$ by $P(x) = \sum_{k=0}^n \alpha_k x^k$ with degree $n$ (see [Vlcek 2012] for further properties of the polynomial), then the two-layer non-linear network can be approximated by $$ f(x) = W_2 \sigma(W_1 x) \approx W_2 \sum_{k=0}^n \alpha_k (W_1 x)^k, $$ where the power of a vector is applied entry-wise. Under the polynomial representation of the network, the loss function and the variance of the (stochastic) gradient are also polynomial in $w_t$, which is again captured by our theorems about any multiplication term of model weights and stochastic gradient estimators. By this approach our theorems should extend to non-linear networks.
>
> Another line of attack would be to show similar results with the help of the existing neural tangent kernel results under the over-parameterized setting. Again, the neural network can be approximated by linear models and the theorems in this paper can be applied.
>
> We are working on the above two ideas but it takes tens of pages mathematical rigor to prove them. Therefore, the authors decide to handle them separately and leave this paper as the initial results in dealing with ``$t$-step'' quantities in deep learning.
>
> For non-Gaussian samples with general networks, we can approximate the dynamics of SGD as a stochastic differential equation \begin{align*}
>     \mathrm{d} w(t) = - \alpha(t) g(w(t)) + \frac{\alpha(t)}{\sqrt{b}} C(t)^{\frac{1}{2}} \mathrm{d}W(t),
> \end{align*}
> where $g(\cdot)$ is the stochastic gradient function, $\alpha(t)$ is the learning rate, $C(t)^{\frac{1}{2}}$ is the factorization of the covariance matrix of $g(w(t))$, and $\mathrm{d}W(t)$ is noise. Taking $b \rightarrow \infty$, the second term in the above equation goes to 0, and we end up with a simple SDE where an analytical solution exists. By approximating this solution with polynomials and plugging it back to the equation, we should be able to give a polynomial representation of $w(t)$. Then the variance of $g(w(t))$ can also be written as a polynomial and it align with Theorem 4 in the paper.
>
> In the heavily over-parameterized case with ReLU networks, we can show similar results with the help of the existing neural tangent kernel results. However, we do not want to make assumptions on the network capacity as we try to make the results as general as possible. Deep linear networks are models where we do not need any assumption on the model width/depth, so we spend most of the paper discussing them.
>
> ## References
>
> - Jacot, Arthur, Franck Gabriel, and Clément Hongler. "Neural tangent kernel: Convergence and generalization in neural networks." Advances in neural information processing systems. 2018.
>
> - Du, Simon S., et al. "Gradient descent provably optimizes over-parameterized neural networks." arXiv preprint arXiv:1810.02054 (2018).
>
> - Arora, Sanjeev, et al. "Fine-grained analysis of optimization and generalization for overparameterized two-layer neural networks." arXiv preprint arXiv:1901.08584 (2019).
>
> - Lee, Jaehoon, et al. "Wide neural networks of any depth evolve as linear models under gradient descent." Advances in neural information processing systems. 2019.
>
> - Kawaguchi, Kenji. "Deep learning without poor local minima." Advances in neural information processing systems. 2016.
>
> - Arora, Sanjeev, et al. "A convergence analysis of gradient descent for deep linear neural networks." arXiv preprint arXiv:1810.02281 (2018).
>
> - Laurent, Thomas, and James Brecht. "Deep linear networks with arbitrary loss: All local minima are global." International conference on machine learning. PMLR, 2018.
>
> - Bottou, Léon, Frank E. Curtis, and Jorge Nocedal. "Optimization methods for large-scale machine learning." Siam Review 60.2 (2018): 223-311.
>
> - Vlcek, Miroslav. "Chebyshev polynomial approximation for activation sigmoid function." Neural Network World 4.12 (2012): 387-393.

---

### Decision · Program_Chairs · 2021-01-10
**Final Decision**

**Decision:**

Reject

**Comment:**

This work analyses the impact of mini-batch size on the variance of the gradients during SGD, in the context of linear models. It shows an inverse relationship between the variance of the gradient and the batch size for such models, under certain assumptions. Reviewers generally agree that the work is theoretically sound. However, all reviewers believe that the contributions of this work are limited. This concern was not adequately addressed during the discussion phase and led to the ultimate decison to reject.